# The Role of Bruton’s Kinase Inhibitors in Chronic Lymphocytic Leukemia: Current Status and Future Directions

**DOI:** 10.3390/cancers14030771

**Published:** 2022-02-02

**Authors:** Tadeusz Robak, Magda Witkowska, Piotr Smolewski

**Affiliations:** 1Department of Hematology, Medical University of Lodz, 93-510 Lodz, Poland; 2Department of Experimental Hematology, Medical University of Lodz, 93-510 Lodz, Poland; magdalena.witkowska@umed.lodz.pl (M.W.); piotr.smolewski@umed.lodz.pl (P.S.)

**Keywords:** acalabrutinib, BTK, CLL, COVID-19, ibrutinib, DTRMWXHS-12, fenebrutinib, nemtabrutinib, orelabrutinib, pirtobrutinib, spebrutinib, TG-1701, tirabrutinib, zanubrutinib

## Abstract

**Simple Summary:**

The availability of Bruton’s tyrosine kinase (BTK) inhibitor ibrutinib has undoubtedly reshaped the initial management of chronic lymphocytic leukemia (CLL). Acalabrutinib and zanubrutinib are selective second-generation BTK inhibitors designed to have high specificity for BTK and minimize off-target effects. However, despite the positive impact of these drugs on patient outcomes, their introduction has created new practical challenges for clinicians, mainly due to their adverse events and the development of drug resistance. Therefore, new combinations of BTK inhibitors and their combinations are currently being tested. This review summarizes new data about the approved drugs and the agents in clinical development for therapeutic use in CLL.

**Abstract:**

The use of Bruton’s tyrosine kinase (BTK) inhibitors has changed the management and clinical history of patients with chronic lymphocytic leukemia (CLL). BTK is a critical molecule that interconnects B-cell antigen receptor (BCR) signaling. BTKis are classified into two categories: irreversible (covalent) inhibitors and reversible (non-covalent) inhibitors. Ibrutinib was the first irreversible BTK inhibitor approved by the U.S. Food and Drug Administration in 2013 as a breakthrough therapy in CLL patients. Subsequently, several studies have evaluated the efficacy and safety of new agents with reduced toxicity when compared with ibrutinib. Two other irreversible, second-generation BTK inhibitors, acalabrutinib and zanubrutinib, were developed to reduce ibrutinib-mediated adverse effects. Additionally, new reversible BTK inhibitors are currently under development in early-phase studies to improve their activity and to diminish adverse effects. This review summarizes the pharmacology, clinical efficacy, safety, dosing, and drug–drug interactions associated with the treatment of CLL with BTK inhibitors and examines their further implications.

## 1. Introduction

Chronic lymphocytic leukemia (CLL) is an indolent form of small mature B-cell leukemia that predominantly affects older individuals [1,2]. It is the most common type of leukemia in Western countries, with a median age at diagnosis of 72 years. CLL is responsible for approximately 40% of all adult leukemias and 11% of all hematological malignancies. The incidence of the disease is 3.5 cases per 100,000 inhabitants per year [3]. In addition, the World Health Organization (WHO) also recognizes small lymphocytic lymphoma (SLL) as a disease entity. Its cells exhibit the same immunophenotype (CD5+/CD19+ and CD5+/CD23+) and morphology as in CLL but does not primarily involve the bone marrow; as such, small lymphocytic lymphoma (SLL) is classified as a combined lymphoproliferative disease, i.e., as SLL/CLL [4]. Consequently, many clinical trials have considered CLL and SLL in the same group of malignancies. In a recent study, relative survival improved significantly for CLL patients diagnosed between 1985 and 2015 [4]. However, despite these findings and recent changes in therapeutic strategies in CLL, the disease remains incurable in the majority of patients. Most of the patients will relapse sooner or later, and some will be resistant to the treatment available. Bruton’s tyrosine kinase (BTK) is a member of the TEC kinase family [5]. It plays an important role in malignant B lymphocyte proliferation and survival in CLL [6]. BTK inhibitors (BTKis) have become very important targets in the treatment of inflammatory reactions and autoimmune diseases, as well as of B-cell malignancies, including CLL. These agents have transformed CLL management in both previously untreated and relapsed/refractory patients [7,8,9,10,11,12]. BTK inhibitors are classified into two categories: irreversible (covalent) inhibitors and reversible (non-covalent) inhibitors (Table 1) [13]. Ibrutinb, acalabrutinib, and zanubrutinib are irreversible BTKis, binding covalently to the Cys481 residue in the ATP binding pocket of BTK [14]. However, most patients develop resistance to treatment with currently approved BTK inhibitors, and novel therapies are urgently needed. Ibrutinib is a first-in-class, irreversible, oral, once-daily BTKi approved for the treatment of CLL/SLL. As a single agent, ibrutinib has led to prolonged progression-free survival (PFS) and overall survival (OS) in patients with previously treated CLL [15]. Of the remaining irreversible BTKis, acalabrutinib and zanubrutinib are the most advanced in clinical trials. Both have been investigated in randomized clinical trials in CLL patients, some of which have made direct comparisons with ibrutinib [16,17]. 

Considerable efforts are underway to reduce the toxicity of CLL treatments and improve their efficacy, which have resulted in the design of a number of new agents. However, while these agents have shown encouraging results in early clinical results, no long-term data are yet available. This review therefore summarizes the pharmacology, clinical efficacy, safety, dosing, and drug–drug interactions of BTKis in the treatment of CLL and discusses their further implications. It includes the most recent results from the ongoing clinical trials and preclinical studies. It also presents novel concepts for the management of CLL, including the use of optimal drug combinations, novel irreversible and reversible BTKis, and more precise and individualized approaches intended to enhance the progress and development of well-designed clinical trials.

## 2. Mechanism of Action 

B-cell receptor (BCR) signaling is an essential component of the development and survival of normal and malignant B cells [18]. In a CLL-like mouse model, BTK deficiency significantly delayed development and reduced leukemia infiltration but still resulted in the development of lymphogenesis. In vivo BTK inhibition abolished tumor formation, while BTK overexpression increased cancer incidence and overall mortality [19]. 

The BCR signaling pathway includes several elements, the most important of which is BTK. This kinase consists of several regions, including the pleckstrin homology (PH) domain, the Tec homology (TH) domain, the Src homology (SH3) domain, the SH2 domain, and the C-terminal region with kinase activity. BCR signaling can be tonic or chronically activated. It is initiated by the antigen binding to the surface immunoglobulin, followed by autophosphorylation of the CD79A/CD79B heterodimer by kinases from Src family [20]. It has been also demonstrated that the Src kinase Lyn plays a pivotal role in the pathogenesis of CLL, mainly due to its constitutive phosphorylation. In CLL cells, constitutive phosphorylation of phospholipase C-γ2 (PLC-γ2), spleen tyrosine kinase (Syk), protein kinase C (PKC)-β, BTK, and phosphoinositide 3′-kinase (PI3K) resulted in the activation of the nuclear factor kappa B (NF-κB) pathway [21]. This process may interact with microenvironmental stimuli and thus initiate the maintenance of survival, proliferation, or migration of CLL cells [6,11]. Targeting the BCR pathway via the inhibition of BTK has evolved the treatment of some B-cell malignancies, including CLL. Most BTKis target the ATP binding site of BTK, which contains cysteine (C481), known to target several irreversible inhibitors. C481 in BTK can act as a nucleophile and form a covalent bond with the inhibitor. 

BTK inhibitors are the most advanced targeted drugs in B-cell lymphoid malignancies. Their characteristics are presented in Table 1. The majority of currently approved BTKis are irreversible inhibitors. Three of them, ibrutinib, acalabrutinib, and zanubrutinib, have been approved by the U.S. Food and Drug Administration (FDA) for the treatment of B cell neoplasms and graft-versus-host disease (GVHD). 

These three approved BTKis have been found to demonstrate certain similarities. All irreversibly covalently bind to cysteine 481 in the ATP binding pocket of BTK. Biochemical binding kinetics indicate that ibrutinib is the most potent BTKi of the three, followed by zanubrutinib and acalabrutinib, but differences in biochemical potency were partially obscured in cellular assays using human peripheral blood mononuclear cells or human white blood cells (all below 10 nM). Of the three drugs, the highest selectivity and the lowest off-target ratio were demonstrated by acalabrutinib [22]. These observed differences will of course influence the dosage, efficacy, and side effects of their use in clinical practice. For example, when administered once a day, acalabrutinib had a shorter half-life than ibrutinib; in addition, twice-daily dosing yielded higher BTK load than once-daily dosing (95.3% vs. 87.6%). These results suggest that acalabrutinib requires two doses per 24 h [23]. Moreover, acalabrutinib was found to have an inhibitory effect on epidermal growth factor receptor (EGFR), which can be associated with rash and severe diarrhea [24]. Additionally, ibrutinib can trigger TEC kinase, which contributes to platelet dysfunction and increases the risk of bleeding [25]. BTK inhibitors can be combined with other targeted agents known to be active in CLL. They have demonstrated synergistic effects with anti-CD20 monoclonal antibodies in inducing apoptosis in tumor cells [26,27]. In addition, the combination of ibrutinib and a BCL-2 antagonist showed additive or more than additive cytotoxicity in vitro against CLL cells from patients treated with ibrutinib [28]. These findings suggest that combinations of BTK inhibitors, BCL-2 antagonists, and/or anti-CD20 monoclonal antibodies should be tested clinically against CLL to increase antileukemic efficacy and reduce the risk of acquired resistance. The mechanism of action of BTK inhibitors is summarized in Figure 1. 

## 3. Irreversible Covalent BTK Inhibitors

### 3.1. Ibrutnib

Ibrutnib (PCYC-1102, Imbruvica^®^, Pharmacyclics LLC, Sunnyvale, CA, USA) was discovered in 2007 as an irreversible inhibitor for BTK [29]. The drug was approved by the FDA in 2013 for the treatment of mantle cell lymphoma (MCL) and in 2014 for relapsed/refractory CLL. Subsequently, ibrutinib was approved for Waldenström’s macroglobulinemia (WM), GVHD, and marginal zone lymphoma (MZL). Ibrutinb binds to the cysteine 481 (C481) residue of BTK, irreversibly inhibiting phosphorylation of downstream kinases in the BCR signaling pathway and blocking B-cell activation. However, ibrutinib can also block other kinases, including EGFR, ErbB2, ITK, and TEC (Figure 1) [30]. Ibrutinib treatment can result in adverse events (AEs) such as bleeding and cardiac arrhythmia due to its off-target activity [31]. 

#### 3.1.1. Ibrutinib in Relapsed and Refractory CLL

Several clinical studies with ibrutinib in monotherapy have been performed in patients with CLL [32]. An initial phase 1b/2 PCYC-1102 study and PCYC-1103 extension study included patients receiving single-agent ibrutinib in first-line or relapsed/refractory (R/R) CLL/SLL. Ibrutinib was administered in a daily dose of 420 or 840 mg until unacceptable toxicity or disease progression. The final analysis yielded similar results in overall response rate (ORR) between first-line patients (87%) and relapsed/refractory (R/R) patients (89%); however, the first-line group demonstrated higher complete response (CR), i.e., 35%, than the R/R group (10%) [33]. The estimated 7-year progression-free survival (PFS) rates were 83% in the first-line group and 34% in R/R patients. Forty-one patients had CLL progression, including 11 with Richter’s transformation. Median PFS was not reached (NR) with first-line ibrutinib. In R/R CLL/SLL, patients with 17p deletion (del(17p)) demonstrated an overall median PFS of 52 months and 26 months. Estimated 7-year overall survival (OS) rates were 84% in previously untreated patients and 55% in R/R patients. Most common grade ≥3 adverse events (AE) were hypertension (28%), pneumonia (24%), and neutropenia (18%). Importantly, grade ≥3 AEs declined over time in most patients, excluding hypertension The results of the long follow-up of this study indicate that ibrutinib, as a single agent, can induce sustained responses and long-term tolerability in previously untreated and R/R CLL patients.

The phase 3 RESONATE trial compared ibrutinib with ofatumumab in 391 patients with R/R CLL (Table 2). The median follow-up in this study was 65.3 months. Ibrutinib significantly improved both PFS and OS. Among the ibrutinib-treated patients, ORR was 91%, including 11% CR. The median PFS was 44.1 months for the ibrutinib arm and 8.1 months for the ofatumumab arm (*p* <0.001). Patients receiving ibrutinib showed longer OS than those treated with ofatumumab (HR: 0.639; 95% CI: 0.418–0.975). Ibrutinib was found to be a relatively safe drug with a good toxicity profile. At a median follow-up of 41 months, hypertension was observed in 21% of patients treated with ibrutinib and atrial fibrillation in 12%; in addition, grade 3 and higher AEs were found in 9% and 6%, respectively. Ibrutinib was discontinued due to adverse events in 16% of patients [33]. The study data indicate that treatment with ibrutinib resulted in a greater improvement in disease symptoms and hematological parameters compared to ofatumumab and significantly improved the quality of life of patients with CLL/SLL. The long-term effects of ibrutinib treatment among patients taking part in the RESONATE study [34], i.e., the longest follow-up analysis so far for an orally administered target inhibitor in CLL, with a median follow-up period of 65.3 months, were recently published. It was found that patients receiving ibrutinib demonstrated significantly longer median PFS in that those with ofatumumab (44.1 months vs. 8.1 months). In addition, this favorable PFS was also noted for ibrutinib treatment in a high-risk population with TP53 mutation, del(17p) and del (11q), but with *IgHV* unmutated status. 

In an attempt to improve the efficacy of ibrutinib treatment in R/R CLL, studies have examined its efficacy in combination with rituximab, ofatumumab, venetoclax, or rituximab and bendamustine (BR) [35,36,37]. Ibrutinib alone was compared with ibrutinib with rituximab in a trial encompassing 208 patients with CLL and 181 with recurrent CLL. In addition, 27 patients were treatment-naïve with high-risk genetic factors: (del)17p or *TP53* mutation [38]. The addition of rituximab to ibrutinib was not found to improve PFS in relapsed and untreated high-risk CLL patients.

Elsewhere, the combination of ibrutinib with BR was found to be more effective than with BR alone: the ibrutinib plus BR (87.2%) group demonstrated a significantly higher ORR than the BR (66.4%) group alone, with continuous improvement seen over time with the ibrutinib arm [39], while the ibrutinib + BR arm also demonstrated longer PFS.

The three-year PFS was 70.2% of patients in the ibrutinib + BR arm but only 15.5% in the BR monotherapy arm [39]. Ibrutinib and ibrutinib + BR therapy demonstrated similar PFS and OS results; however, it is not known whether the deeper responses observed for ibrutinib + BR will translate into improved OS and PFS [40].

**Table 2 cancers-14-00771-t002:** Phase 3 clinical trials of BTK inhibitors in relapsed/refractory chronic lymphocytic leukemia.

Study	Treatment	Patients, N	Median Age [years]	Median Follow-Up [months]	ORR (CR)	Median PFS	Median OS	Discontinu Ation Rates
Byrd et al. [33] Munir et al. [34] (RESONATE)	Ibrutinib vs. Ofatumumab	195 vs. 196	67 vs. 67	41	91% (21%) vs. 82% (11%)	44.1 m vs. 8.1 m (*p* ˂ 0.0001)	67.7 m vs. 65.1 m HR = 0.810	16% for ibrutinib
Fraser et al. [39] (HELIOS)	Ibrutinib + BR vs. BR	289 vs. 289	64 vs. 63	34.8	87% (38%) vs. 66% (8%)	NR vs. 13.3 m, *p* < 0.0001	NR vs. NR, *p* = 0.0598	24% vs. 45%
Sharman et al. [41] (GENUINE)	Ibrutinib + Ublituximab vs. Ibrutinib	64 vs. 62	66 vs. 67	41.6	83% (17%) vs. 65% (3%)	NR vs. 35·9 m *p* = 0.016	NR vs. NR, *p* = 0.12	15% vs. 12%
Byrd et al. [16] (ELEVATE RR)	Acalabrutinib vs. Ibrutinib	268 vs. 265	66 vs. 65	31.3	81.0% (NA) vs. 77% (NA)	38.4 vs. 38.4 *p* ˂ 0.001	NR vs. NR HR, 0.82	14.7% vs. 21.3%
Ghia et al. [42] (ASCEND)	Acalabrutinib vs. IR or BR	155 vs. 155	65 vs. 67	40.9	81% (0%) vs. 75% (1%)	NR vs. 16.5 m *p* < 0.0001	89% vs. 80% at 1 y *p* < 0.0001).	14.7% vs. 21%
Hillmen et al. [17] ALPINE	Zanubrutinib vs. Ibrutinib	Total 415	66 vs. 65	15	76% (NA) vs. 64%(NA)	94.9% vs. 84% at 1 y *p* < 0.0007	97% vs. 93% at 1 y	7.8% vs. 13.0%

Abbreviations: BR—Bendamustine and rituximab, FCR—fludarabine, cyclophosphamide and rituximab, HR—hazard ratio, IR—idelalisib + rituximab, m—months, N—number of patients, CR—complete remission, PFS—progression free survival, R—rituximab, NA—not available, NR—not reached, O—obinutuzumab, OS—overall survival, R—rituximab, y—year.

#### 3.1.2. Ibrutinib in Previously Untreated CLL 

In the RESONATE-2 study, 269 patients with untreated CLL patients 65 years or older without del (17p) were randomized to two arms: ibrutinib 420 mg orally daily until disease progression or chlorambucil monotherapy (Table 3) [43]. With a median follow-up of 60 months, PFS was 70% for ibrutinib vs. 12% for chlorambucil (HR (95% CI): 0.146 (0.098–0.218)). As a secondary endpoint, ibrutinib also improved 5-year OS compared to chlorambucil (83% vs. 68%; HR 0.45, 95% CI = 0.266–0.761). The benefits obtained from ibrutinib treatment were also observed in patients with TP53 mutation, 11q deletion, and/or unmutated *IgHV* (PFS: HR (95% CI): 0.083 (0.047–0.145); OS: HR (95% CI): 0.366 (0.181–0.736). ORR was 92% with ibrutinib (CR, 30%; 11% at primary analysis). Most common grade ≥3 AEs included neutropenia (13%), anemia (7%), pneumonia (12%), hypertension (8%), and hyponatremia (6%). This study demonstrated that, with long follow-up, most patients with high-risk CLL could receive continuous treatment with rituximab used as a single agent, with sustained response and acceptable tolerability.

Subsequent studies compared ibrutinib, alone or in combination with CD20 monoclonal antibodies, with standard immunochemotherapy regimens (Table 3). In the iLLUMINATE study, previously untreated patients aged 65 years or older and younger patients with coexisting comorbidities were randomized to receive either obinutuzumab + ibrutinib or obinutuzumab + chlorambucil [44]. Overall response was higher (88%) in the ibrutinib plus obinutuzumab group than in the chlorambucil plus obinutuzumab group (73%). The prevalence of undetectable MRD in BM or PB was 35% and 25% in the respective groups. PFS was significantly longer in the obinutuzumab + ibrutinib arm compared to the obinutuzumab + chlorambucil arm (NR vs. 19 months; HR 0.23, 95% CI = 0.15–0.37, *p* < 0.001). In a subgroup analysis, patients with high-risk characteristics of CLL (del17p, del11q, TP53 mutations, or unmutated IgHV) also showed significant improvement in PFS in the ibrutinib arm compared to immunochemotherapy (median NR vs. 14.7 months; *p* < 0.001). The recently published results from the final analysis, with a median follow-up of 45 months [8], confirmed that the combination of ibrutinib with obinutuzumab is an effective chemotherapy-free regimen, and that is associated with a lower risk of disease progression compared to chlorambucil and obinutuzumab in patients with high-risk disease. Therapy was well tolerated, and no new safety issues were observed. 

Elsewhere (A041202), 547 elderly patients (≥65 years of age) with previously untreated CLL were randomized to receive ibrutinib alone, ibrutinib combined with rituximab, or BR [45]. The 2-year PFS was higher for both ibrutinib alone (87%) and for ibrutinib + rituximab (88%) compared to BR (74%) (*p* < 0.001 for both comparisons). The ORR values for ibrutinib, ibrutinib and rituximab, and BR were 93%, 94%, and 81%, respectively. Among older patients with untreated CLL, treatment with ibrutinib, alone or in combination, offered better PFS compared to BR. However, no significant differences in ORR were observed between those three study groups, as counted after a median follow-up of 38 months. 

**Table 3 cancers-14-00771-t003:** Phase 3 clinical trials of BTK inhibitors in treatment naïve CLL.

Study	Treatment	N	Median Age [years]	Median Follow-Up [months]	ORR (CR) Rate	Median PFS	Median OS	Discontinuation Rates
Burger at al. [9,34] RESONATE 2	Ibrutinib vs. Chlorambucil	136 vs. 133	73 vs. 72	18.4	86% (4%) vs. 35% (2%)	NR vs. 18.9 m; *p* < 0.001	HR 0.16 (95% CI, 0.05–0.56) *p* = 0.001	13% for Ibrutinib
Fraser et al. [39] (HELIOS)	Ibrutinib + BR vs. BR	289 vs. 289	64 vs. 63	63.7	87% (38%) vs. 66% (8%)	NR versus 13.3 m *p* < 0.0001	NR vs. NR, *p* = 0.0598	47% vs. 51.2%
Burger et al. [43]	Ibrutinib vs. Ibrutinib + R	104 vs. 104	65 vs. 65	36	92.3% (20.2%) vs. 92.3% (26.3%)	86% vs 86.9% at 3 y *p* = 0.912)	92% vs. 89% at 3y *p* = 0.572	Ibrutinib 34% at 3 y
Moreno et al. [8,44] (iLLUMINATE)	Ibrutinib + O vs. Chlorambucil + O	113 vs. 116	70 vs. 72	31.3	88% (19%) vs. 73% (9%)	NR vs. 19 m; *p* < 0.00001	86% vs. 85% at 30 m	9% vs. 13%
Woyach et al. [45] (A041202)	BR vs. Ibrutinib vs. Ibrutinib + R.	183 vs. 182 vs. 182	70 vs. 71 vs. 71	38 m	81% vs. 93% vs. 94% at 2 y	74% vs. 87% vs. 88% at 2 y *p* < 0.001	95% vs. 90% vs. 94% at 2 y *p* ≥ 0.65	Ibrutinib 37% Ibrutinib + R 36% at median 38 m
Shanefelt et al. [46] (ECOG 1912)	Ibrutinib vs. FCR	354 vs. 175	56.7 vs. 56.7	33.6	95.8% (17.2%) vs. 81.1% (30.3%)	89.4 vs. 72.9% at 3-yrs *p* < 0.001	98,8% vs. 91.5% at 3-yrs *p* < 0.001	Ibrutinib arm 21.2%
Kater et al. [47] (GLOWE)	Ibrutinib + Venetoclax vs. Chlorambucil + O	106 vs. 105	71 vs. 71	27.7	86.8% (38.7%) vs. 84.8% (11.4%)	NR vs. 21 m *p* < 0.0001	11 deaths vs. 12 deaths HR 1.048	NA
Tam et al. [48] (SEQUOIA)	Zanubrutinib vs. BR	241 vs. 238	70 vs. 70	26.2	94.6% (6.6%) vs. 85.3% (15.1%)	85.5% vs. 69.5% at 2 y *p* < 0.0001	94.3 vs. 94.6 at 2 y	8.3% vs. 13.7%
Sharman et al. [49,50] (ELEVATE TN)	Acalalabrutinib vs. Acalalabrutinib + O vs. Chlorambucil + O	179 vs. 179 vs. 177	70 vs. 70	46.9	89.9% (11.2%) vs. 96.1% (30.7%) vs. 87.4% (13.0%)	NR vs. NR vs. 27.8 m *p* < 0.0001	88% vs. 92.9% vs. 92% at 4 y *p* = 0.0836	Acalabrutinib 30.7% vs. Acalabrutinib + O 25.1% vs. Chlorambucil + O 22.6%

Abbreviations: BR—Bendamustine and rituximab, CR—complete remission, FCR—fluorouracil, cyclophosphamide and rituximab, m—months, HR—hazard ratio, N- number of patients, NA—not available, NR—not reached, O—obinutuzumab, ORR—overall response rate, OS—overall survival, PFS—progression free survival, R—rituximab, FCR—rituximab, fludarabine, cyclophosphamide, y—year.

Shanefelt et al. compared the efficacy and safety of ibrutinib and rituximab with standard chemoimmunotherapy with fludarabine, cyclophosphamide, and rituximab (FCR) in patients with previously untreated CLL, aged 70 years or younger (E1912 study) [46]. Patients received ibrutinib 420 mg orally once daily until disease progression or toxicity and rituximab once every 28 days for six doses. Patients with del (17p) were excluded due to the historically poor response of this population to FCR. The primary endpoint was PFS, and OS was a secondary endpoint. A total of 529 patients were randomized in this study. More patients achieved 3-year PFS with ibrutinib and rituximab compared to FCR (89.4% vs. 72.9%; *p* < 0.001). Ibrutinib combined with rituximab yielded better 3-year OS than FCR (98.8% vs. 91.5%; *p* < 0.001). Both arms demonstrated similar rates of grade ≥3 AEs (80.1% vs. 79.7%). These findings confirm that combined ibrutinib + rituximab therapy offers better PFS and OS than FCR in fit/younger previously untreated CLL patients. 

Another study compared the effectiveness of ibrutinib combined with ublituximab, another anti-CD20 monoclonal antibody, which was also compared with ibrutinib alone in R/R CLL patients with at least one of 17p deletion, 11q deletion, or TP53 mutation [41]. Ublituximab with ibrutinib induced a higher OR rate to ibrutinib alone in this high-risk patient population, but with a similar safety profile. In addition, ublituximab plus ibrutinib treatment was associated with greater minimal residual disease (MRD) negativity in peripheral blood (PB), bone marrow (BM), or both (42%) compared to ibrutinib (6%, *p* < 0.0001). 

Finally, fixed-duration treatment with ibrutinib and venetoclax was investigated in a phase 2 study in R/R patients (CLARITY), and in previously untreated patients (CAPTIVATE) [35,51,52]. In the CLARITY study, the primary end point was the eradication of minimal residual disease (MRD) after 12 months of treatment with ibrutinib and venetoclax [35]. Therapy was given for a limited period and then stopped if patients achieved deep remission with MRD negativity. Undetectable MRD was defined as fewer than one CLL cell in 10,000 leukocytes. Out of 53 evaluated patients, MRD negativity was achieved in 28 (53%) in the peripheral blood (PB) and in 19 (36%) in the bone marrow (BM) after 12 months, with an OR rate of 89% and CR 51%. After a median follow-up of 21.1 months, all patients were alive, and only one patient demonstrated progression. This is the first study to demonstrate that a combination of two drugs targeting the key pathogenetic pathways in CLL induced MRD-negative responses in a significant number of patients. 

In the CAPTIVATE study, 159 older/unfit previously untreated patients with CLL/SLL received three cycles of ibrutinib (420 mg/d orally) and, subsequently, 12 cycles of ibrutinib and venetoclax. ORR was 96%, and CR was achieved by 88 patients (55%), including 78 (89%) patients with CR duration of one year or longer. Undetectable MRD was achieved in 77% of patients in PB and 60% of patients in BM. A maintained response after two years was observed in most patients (90%) compared with 41% for chlorambucil plus obinutuzumab. Two-year PFS was 95%, and two-year OS was 98%. The most common grade 3/4 AEs were neutropenia (33%), hypertension (6%), and decreased neutrophil count (5%). Fixed-duration ibrutinib–venetoclax combination therapy induced deeper and better sustained responses than chlorambucil–obinutuzumab in previously untreated older/unfit patients with CLL. 

In the GLOW open-label, randomized phase 3 study, 106 previously untreated patients were randomized to receive ibrutinib plus venetoclax and 105 to receive chlorambucil plus obinutuzumab (Table 3) [47]. Patients with del(17p) or *TP53* mutations were excluded. The participants received 420 mg/day ibrutinib for three months, followed by 12 cycles of ibrutinib plus venetoclax. In the control arm, the patients received six cycles of chlorambucil plus obinutuzumab at standard doses. The independent review committee (IRC) found ORR values of 86.6% for the ibrutinib–venetoclax arm and 84.8% for the chlorambucil–obinutuzumab arm. However, the ibrutinib–venetoclax arm yielded a significantly higher CR rate (38.7%) than chlorambucil–obinutuzumab (11.4%) (*p* < 0.0001). The undetectable MRD in BM scores at three months following end of treatment were 51.9% and 17.1%, respectively (*p* < 0.0001). With a median follow-up of 27.7 (range, 1.7 to 33.8) months, median IRC-assessed PFS was not reached for the ibrutinib–venetoclax arm and 21.0 months for the chlorambucil–obinutuzumab arm; estimated 24-month PFS rates were 84.4% versus 44.1%, respectively. In addition, 90% of responders to ibrutinub–venetoclax demonstrated a maintained response after two years compared with 41% of responders to chlorambucil–obinutuzumab.

### 3.2. Acalabrutinib 

Acalabrutinib (ACP-196, Calquence^®^, AstraZeneca Pharmaceuticals LP) is a novel second-generation oral, potent, highly selective, covalent BTKi, designed by Acerta Pharma. It is possible for acalabrutinib to covalently bind to the C481 residue in BTK via a reactive butinamide group. The new compound demonstrates different properties of ibrutinib, which reduce off-target binding: for example, acalabrutinib does not inhibit EGFR or ITK. 

In the phase 1/2 trial, acalabrutinib was evaluated in 61 patients with relapsed/refractory CLL. The patients received acalabrutinib at a dose of 100 to 400 mg once daily in the dose-escalation phase (Phase 1) and 100 mg twice daily in the dose-escalation part (phase 2) [53]. The patients had received a median of three lines of previous treatments for CLL. Among high-risk patients, 31% had chromosome 17p13.1 deletion, and 75% had unmutated *IGVH*. At a median follow-up of 14.3 months, the OR rate was 95%, including 85% with a partial response (PR) and 10% with a PR with lymphocytosis. All patients (100%) with chromosome 17p13.1 deletion responded to the treatment. The most common AEs were headache (43%), diarrhea (39%), and increased weight (26%), and most were of grade 1 or 2. 

The results of the updated and expanded phase 2 study confirmed the efficacy and long-term safety of acalabrutinib in previously treated patients with CLL from the phase 1b/2 study. In a recent report, 134 patients were treated with 100 mg acalabrutinib twice daily for a median of 41 months [54]; the ORR was 94% with similar responses demonstrated by patients with del(11)(q22.3), del(17)(p13.1), complex karyotype, or *IGVH* mutation status. Median PFS was not reached (NR) and estimated 45-month PFS was 62%. The most commonly seen AEs were diarrhea (52%) and headache (51%). All grades of atrial fibrillation were observed in 7% of patients and major bleeding AEs in 5%. Importantly, 56% of patients remain on treatment. The most common reasons for discontinuing therapy were progressive disease (21%) and AEs (11%). 

Acalabrutinib was also evaluated in a group of 99 treatment-naïve CLL patients, in a single-arm phase 1/2 trial (ACE-CL-001) [55]. Among the recruited patients, 57 (62%) had unmutated *IGVH*, and 12 (18%) had TP53 mutation or deletion. Acalabrutinib was given orally 200 mg once daily, or 100 mg twice daily until progression or unacceptable toxicity. After a median follow-up of 53 months, 14 patients had discontinued treatment, and 85 remained on treatment. Discontinuation was due to AEs in six patients and disease progression in three. The OR rate was 97%, including 90% PR and 7% CR. Serious AEs were reported in 38 patients (38%). Grade ≥3 events of special interest were infection (15%), hypertension (11%), bleeding events (3%), and atrial fibrillation (2%). These studies confirmed the high efficacy, durable responses, and good safety profile of acalabrutinib in previously untreated and R/R patients.

In the subsequent studies, acalabrutinib combined with obinutuzumab was investigated in phase 1b/2 and phase 3 trials. In a phase 1b/2 study, Woyach et al. assessed combined acalabrutinib + obinutuzumab therapy in 19 treatment-naïve and 26 R/R CLL patients [56]. The regimen consisted of 100 mg acalabrutinib twice daily until progression, with obinutuzumab ramped from 100 to 1000 mg/day for up to six cycles. ORR was observed in 95% of treatment-naïve patients and 92% of R/R patients. CR was reached in 32% of naïve and 8% of R/R patients. In addition, 94% previously untreated and 88% R/R patients were progression free at three years. Grade 3/4 AEs were reported in 71% of patients. The phase 3 ELEVATE TN (NCT02475681) study evaluated acalabrutinib treatment in previously untreated CLL patients (Table 3) [49,50]. A total of 535 patients were randomized to acalabrutinib alone, acalabrutinib + obinutuzumab, or chlorambucil + obinutuzumab arms. Of the patients, 63% had unmutated IGHV and 9% del(17p). The patients assigned to an acalabrutinib arm received oral acalabrutinib 100 mg twice a day until progression or intolerable toxicity. At a median follow-up of 46.9 months, treatment was continued in 74.9% of patients in the acalabrutinib–obinutuzumab arm and in 69.3% in acalabrutinib monotherapy arm. Patients treated with acalabrutinib + obinutuzumab had significantly longer PFS compared to chlorambucil + obinutuzumab (median NR vs. 27.8 months; (*p* < 0.0001). Acalabrutinib monotherapy also resulted in a significant increase in PFS compared to chlorambucil + obinutuzumab (median NR vs. 27.8 months; *p* < 0.0001). Median OS was not reached in all groups. The most common grade 3 or higher AE in all three groups was neutropenia: 30% in the acalabrutinib + obinutuzumab group, 9% in the acalabrutinib group, and 41% in the obinutuzumab + chlorambucil group. Infections (grade 3 or higher) were noted in 21% of patients in the acalabrutinib + obinutuzumab arm, 14% in the acalabrutinib monotherapy arm, and 8% in the obinutuzumab + chlorambucil arm. At four-year follow up (median 46.9 months) the acalabrutinib + obinutuzumab arm demonstrated significantly higher ORR (96.1%) and CR (26.8%/) than the obinutuzumab + chlorambucil arm (ORR 82.5%; CR 12.4%/) (*p* < 0.0001). The acalabrutinib monotherapy arm yielded an ORR of 89.9% (*p* = 0.035 vs. obinutuzumab + chlorambucil). The median PFS was NR for acalabrutinib + obinutuzumab and acalabrutinib alone, and 27.8 months for chlorambucil + obinutuzumab (both *p* < 0.0001). Estimated 4-year PFS rates were 87% for acalabrutinib + obinutuzumab, 78% for acalabrutinib, and 25% for chlorambucil + obinutuzumab; however, median OS was NR in all three groups. Estimated 48-month OS rates were 92.9% for acalabrutinib–obinutuzumab, 87.6% for acalabrutinib, and 88.0% for the obinutuzumab–chlorambucil arm. Treatment discontinuation rates were 25.1% in acalabrutinib + obinutuzumab, 30.7% in acalabrutinib alone, and 22.6% in chlorambucil + obinutuzumab. AEs were the most common reasons for treatment discontinuation (12.8%, 12.3%, 14.7%, respectively). These confirm that acalabrutinib, used alone or in combination, is a good therapeutic option in previously untreated CLL patients. 

Acalabrutinib was also investigated in two randomized trials in R/R patients. In the ASCEND multicenter, randomized, open-label phase 3 study, 310 R/R patients were randomized to acalabrutinib monotherapy or the investigator’s choice: idelalisib–rituximab (IR) or bendamustine with rituximab (BR) (Table 2). [42]. In this trial, 78% of patients had unmutated *IGHV*, 16% had del (17p), and 24% had a mutated TP53 gene. The ORR was similar in both arms, including 81% for the acalabrutinib arm and 75% for the investigator’s choice arm (*p* = 0.22). However, at a median follow-up of 16.1 months, acalabrutinib had a significantly higher PFS (83%) than the investigator’s choice arm (56%). Acalabrutinib was more effective across all clinical subgroups with regard to age, sex, ECOG, Rai staging, bulky disease, number of prior therapies, TP53 disruption, *IgHV* status, and complex karyotype. The DOR was not reached with the acalabrutinib arm and was 13.6 months with the investigator’s choice arm. Neither arm reached median OS, with rates exceeding 90%. Grade 3/4 AEs occurred in 45% of patients on acalabrutinib vs. 86% on IR and 43% on BR. The most common high-grade AEs on acalabrutinib were neutropenia, 16% (vs. 40% on IR and 31% on BR); anemia, 12% (vs. 7% on IR and 9% on BR); and pneumonia, 5% (vs. 9% on IR and 3% on BR). Atrial fibrillation (AF) was diagnosed in 5% patients on acalabrutinib and in 3% in the investigator’s choice arm. Bleeding of any grade occurred in 26% (acalabrutinib) and 7% of patients (investigator’s choice). The ASCEND study is an important study, as it is one of the first to compare two different small molecule inhibitors (idelalisib and acalabrutinib) in CLL.

In the second phase 3 trial (ELEVATE RR), acalabrutinib was compared with ibrutinib in 533 patients with R/R CLL (Table 2) [16,57]. This is the first direct comparison of the less-selective BTK inhibitor ibrutinib with the more-selective inhibitor BTK acalabrutinib in CLL. In this study, patients with confirmed del(17) or del(11) received oral acalabrutinib 100 mg twice daily or ibrutinib 420 mg once daily until progression or unacceptable toxicity. At the time of analysis, 124 (46.3%) patients treated with acalabrutinib and 109 (41.1%) patients receiving ibrutinib were on treatment. After a median follow-up of 40.9 months, both groups demonstrated similar PFS (median 38.4 months in both arms). However, the incidence of all-grade atrial fibrillation/atrial flutter was significantly lower with acalabrutinib (9.4%) versus ibrutinib (16.0%; *p* = 0.02). Grade 3 or higher infections were similar (30.8% vs. 30.0%), and Richter transformations (3.8% vs. 4.9%) were comparable between groups. Neither arm reached median OS. Overall, 63 deaths (23.5%) were noted in the acalabrutinib arm, compared to 73 deaths (27.5%) in the ibrutinib arm. In addition, the rate of treatment discontinuations due to adverse events was 14.7% in the acalabrutinib arm and 21.3% in the ibrutinib arm. To summarize, acalabrutinib treatment demonstrated fewer cardiovascular adverse events compared to ibrutinib, with non-inferior PFS, in pretreated, high-risk CLL patients. 

Time-limited treatment with acalabrutinib, venetoclax, and obinutuzumab was also investigated in patients with previously untreated CLL in a phase 2 study [58]. Therapy consisted of acalabrutinib alone for cycle 1, and acalabrutinib with obinutuzumab for six cycles. Venetoclax was used from the beginning of cycle 4 until day 1 of cycle 16 or day 1 of cycle 25. Patients with CR and undetectable MRD in the BM could discontinue therapy at the start of cycle 16 or at the start of cycle 25 if they were at least in PR. Undetectable MRD was defined as less than 1 CLL cell per 10 000 leucocytes, as measured by four-color flow cytometry. At cycle 16, day 1, 14 (38%) of 37 patients obtained CR with undetectable MRD in the BM. An ongoing phase 3 study (NCT03836261) is currently comparing the effects of combined acalabrutinib and venetoclax, with and without obinutuzumab, with the investigator’s choice of chemoimmunotherapy in previously untreated CLL.

Sheng et al. compared acalabrutinib + obinutuzumab, ibrutinib + obinutuzumab, and venetoclax + obinutuzumab in untreated CLL using a network meta-analysis from three trials [59]. In this analysis of 1017 patients, the acalabrutinib + obinutuzumab group demonstrated longer PFS compared with ibrutinib + obinutuzumab (HR:0.43, *p* = 0.02) and venetoclax + obinutuzumab (HR:0.30, *p* < 0.001). Sensitivity analysis also found acalabrutinib + obinutuzumab to have better PFS than ibrutinib + obinutuzumab (HR:0.46, *p* = 0.04) and venetoclax + obinutuzumab (HR:0.34, *p* = 0.002). The investigator assessment found acalabrutinib + obinutuzumab to have the highest probability of obtaining the longest PFS (98.0%). However, this regimen offered no survival advantage compared to ibrutinib + obinutuzumab (HR:0.51, *p* = 0.21) or venetoclax + obinutuzumab (HR:0.38, *p* = 0.07). In addition, no significant difference in AEs analysis was observed; however, prospective clinical trials comparing these regimens are needed to confirm this observation.

In 2017, the U.S. FDA granted acalabrutinib accelerated approval for treating mantle cell lymphoma (MCL) patients after at least one prior course of therapy. Two years later, in November 2019, following two phase 3 clinical trials, this approval was extended to include treatment of adults with CLL/SLL.

### 3.3. Zanubrutinib 

Zanubrutinib (BGB-3111, Brukinsa^®^, BeiGene USA, Inc., San Mateo, CA, USA) is a next-generation irreversible inhibitor of BTK developed by BeiGene in 2012 for the treatment of B-cell malignancies [48,60,61,62,63]. It was designed to offer greater BTK occupancy and lower off-target inhibition of TEC- and EGFR-family kinases. Zanubrutinib demonstrates greater selectivity than ibrutinib for BTK compared to other receptor tyrosine kinases, which may result in a lower incidence of off-target toxicities and reduced severity [48,61]. Moreover, zanubrutinib is similar to acalabrutinib, with less activity on TEC and ITK [62]. Like ibrutinib, zanubrutinib forms H bonds with the residues of the E475 and M477 regions. The drug favorably alters the immune microenvironment by lowering the level of checkpoint molecules on suppressor cells and reducing the number of adhesion/homing receptors on B-cells [63].

In a phase 1/2 study, Tam et al. evaluated 78 patients with CLL/SLL treated with zanubrutinib monotherapy [64]. After a median follow-up of 13.7 months, ORR was 96.2% (75/78), with two patients (2.6%) achieving CR and 63 (80.8%) PR. In approximately 20% of patients, zanubrutinib showed off-target effects, such as diarrhea and rash (all grade 1 or 2 AEs). Subsequently, updated safety and efficacy data, with a median follow-up of 25.1 months, were reported in a larger cohort of CLL/SLL patients [65]. The patients were treated with zanubrutinib at doses ranging from 40 mg once daily to the final phase 2 dose of 160 mg twice daily or 320 mg once daily, until disease progression or unacceptable toxicity. The cohort included 120 efficacy-evaluable patients, including 22 treatment-naïve and 98 R/R patients. ORR was 97% with 14% CR or CR with incomplete bone marrow recovery (CRi). The PFS rate was 97% at one year and 89% at two years; both values were similar in the treatment-naïve and R/R patients. The most common AEs of any grade were contusion (46%), upper respiratory tract infection (39%), diarrhea (30%), cough (28%), headache (23%), fatigue (20%), urinary tract infection (17%), back pain (17%), rash (17%), nausea (16%), and neutropenia (16%). During follow-up treatment, discontinuation was seen in 21/122 patients (17%) mainly due to disease progression (13 patients) and AEs (four patients). Major hemorrhage was observed in 2% of patients and bleeding in 57%. Atrial fibrillation or flutter was observed in three (2.8%). 

The efficacy and safety of zanubrutinib were reported in patients with treatment-naïve CLL/SLL with del(17p): arm C of the SEQUOIA (BGB-3111-304) trial. [66]. In this open-label, global, multicenter, phase 3 study, 109 previously untreated patients were treated with zanubrutinib at a dose of 160 mg twice daily with a median follow-up of 18.2 months (range, 5.0–26.3). At the time of writing, 97 patients (89.0%) were on treatment with zanubrutinib. ORR was 94.5%, including 3.7% CR or CRI. Estimated 18-month PFS was 88.6% and OS 95.1%. The most common AEs included infections (64.2%), bleeding (47.7%; 5.5% grade ≥3 or serious), headache (8.3%), hypertension (8.3%), and myalgia (4.6%). These early results indicate that zanubrutinib is an active and well tolerated BTK inhibitor in previously untreated high-risk CLL patients [66]. 

Recently, interim results for the phase 3 SEQUOIA global registrational trial, arms A and B (BGB-3111-304; NCT03336333), evaluated the efficacy and safety of zanubrutinib vs. BR in previously untreated patients with CLL (Table 3) [48]. Patients without del(17p) received zanubrutinib 160 mg twice daily until disease progression or unacceptable toxicity or BR at the standard doses. ORR was 94.6% for zanubrutinib vs. 85.3% for BR, with respective CR rates of 6.6% vs. 15.1%. At a median follow-up of 26.2 months, zanubrutinib demonstrated longer PFS than BR (HR 0.42, 95% CI 0.28–0.63, 1-sided and 2-sided *p* < 0.0001). Estimated 2-year PFS was 85.5% for zanubrutinib vs. 69.5% for BR, with 2-year OS of 94.3% vs. 94.6%, respectively. Estimated 24-month OS scores were 94.3% for zanubrutinib (95% CI 90.4–96.7%) vs. 94.6% for BR (95% CI 90.6–96.9%). Zanubrutinib was generally well tolerated, with low rates of atrial fibrillation. SEQUOIA is the first randomized, phase 3 study evaluating the efficacy and safety of zanubrutinib in comparison with chemoimmunotherapy in previously untreated CLL patients. The agent demonstrated superior PFS in comparison with BR and a low incidence of cardiac arrhythmia; these findings were similar to those previously reported in patients with chromosome 17p deletion [66].

A phase 3 ALPINE (BGB-3111-305) study was designed to compare zanubrutinib monotherapy with ibrutinib in patients with R/R CLL (Table 2) [17,67]. Endpoints include safety, PFS, duration of response, and OS. An interim analysis from the ALPINE study of zanubrutinib vs. ibrutinib in patients with R/R CLL/SLL were recently presented in abstract form [17]. Patients were randomized to receive zanubrutinib 160 mg twice daily or ibrutinib 420 mg once daily; randomization took place between 5 Nov 2018 and 20 Dec 2019. Both drugs were given until disease progression. At a median follow-up of 15 mo, ORR was significantly higher with zanubrutinib (78.3%) than ibrutinib (62.5%, *p* = 0.0006) In the with the zanubrutinib group, higher ORR scores were observed in patients with del11q (83.6% vs. 69.1%) and del17p (83.3% vs. 53.8%). Overall, 12-month PFS was higher in the zanubrutinib group (94.9%) than the ibrutinib group (84.0%), with respective OS rates of 97.0% and 92.7%. In addition, compared to ibrutinib, the zanubrutinib group demonstrated significantly lower rate of AF/flutter (2.5% vs. 10.1%) (*p* = 0.0014), a lower rate of major bleeding (2.9% vs. 3.9%), and lower rates of AEs leading to discontinuation (7.8% vs. 13.0%). Furthermore, zanubrutinib demonstrated a lower death rate (3.9% vs. 5.8%), higher chance of neutropenia (28.4% vs. 21.7%), and lower rate of grade ≥3 infections (12.7% vs. 17.9%). A phase 2 study evaluating the safety and efficacy of zanubrutinib in ibrutinib-intolerant CLL patients is ongoing (NCT04116437 trial). 

Early results for the SEQUOIA trial (NCT03336333), based on patients with treatment-naive del(17p) CLL/SLL receiving zanubrutinib + venetoclax in arm D, has been recently presented at the American Society of Hematology meeting [68]. With a median follow-up of 9.7 months, 35 patients were included, and 32 patients remained on treatment. Zanubrutinib combined with venetoclax was well tolerated and no new safety signals were identified. For the 31 patients who reached the initial efficacy assessment at three months after starting zanubrutinib, 30 (96.8%) responded to treatment, and one patient had disease progression.

Recently, zanubrutinib combined with obinutuzumab and venetoclax (BOVen) was tested as an initial therapy for 39 treatment-naïve SLL/SLL in a phase 2 trial [69]. The aim of the study was to increase the rates of undetectable minimal residual disease (MRD) and limit treatment duration. Zanubrutinib was administered continuously at 160 mg twice per day throughout the trial. Obinutuzumab was given in cycle 1 at 1000 mg i.v. split between day 1 and 2 (100 mg day 1 and 900 mg day 2) and then at 1000 mg on day 8 and on day 15, while in cycles 2–8, it was only given on day 1. Oral venetoclax was administered in cycle 3, where the dose was ramped up to 400 mg per day starting on day 1 and discontinued after 8–24 cycles if undetectable minimal residual disease (MRD) was achieved in the peripheral blood and bone marrow. Among 39 included patients, 28 (72%) had unmutated *IGVH,* and five (13%) had del (17p) or TP53 mutation. After a median follow-up of 25.8 months, undetectable MRD was identified in both blood and bone marrow in 33 (89%) of 37 patients, and therapy was discontinued after a median of ten cycles. After median treatment of 15.8 months, 31 (94%) of 33 patients had undetectable MRD. The most common AEs of all grades were thrombocytopenia (59%), fatigue (54%), neutropenia (51%), and bruising (51%). The only grade 3 or worse AE was neutropenia, observed in 18% of the patients. These data underline the need for further evaluation of BOVen in CLL patients [69]. Zanubrutinib has received approval by the FDA for the treatment of Waldenstrom macroglobulinemia (WM), R/R MCL, and marginal zone lymphoma.

### 3.4. Other Irreversible BTK Inhibitors

Several other irreversible BTKis have recently been developed and are under clinical investigation in lymphoid malignancies and autoimmune disorders. These include pebrutinib, evobrutinib, olmutinib, tirabrutinib, elsubrutinib (ABBV-105), and tolebrutinib (SAR 442168). 

#### 3.4.1. Spebrutinib 

Similar to ibrutinib, spebrutinib (CC-292, AVL-292, Avila Therapeutics/Celgene) inhibits BTK activity by binding covalently with high affinity to the same cysteine 481 in BTK [70,71]. In preclinical studies, spebrutinib blocked BCR-dependent B-cell activation. In a first-in-human study performed in healthy volunteers, spebrutinib led to near-complete BTK occupancy for 8–24 h (Table 2). Spebrutinibat doses up to 1000 mg/day were investigated in a phase 1 study in 84 patients with R/R CLL/SLL [72,73]. The patients included 21.4% with del(11q), 23.8% with del(17p), and 53.6% with unmutated *IGVH*. CC-292 was well tolerated, with two patients experiencing grade 4 thrombocytopenia, one patient with grade 3 drug-related pneumonitis, and one patient with grade 3 reversible mental-status changes. Common non-hematologic AEs of any grade were diarrhea (68%), fatigue (45%), nausea (35%), cough (27%), pyrexia (27%), and headache (25%). The median response duration was 11.0 months for the 750 mg once daily group and 5.6 months for the 1000 mg once daily, while this value was not yet reached for the 375 mg twice daily and 500 mg twice daily groups. This study indicates that spebrutinib was well tolerated and resulted in dose-dependent responses in R/R CLL/SLL patients, including those with high-risk cytogenetics. However, its clinical activity was lower than that observed in patients with ibrutinib or acalabrutinib.

#### 3.4.2. Orelabrutinib 

Orelabrutinib (ICP-022, Biogen/Innocare Pharma) is an orally available, second-generation BTK inhibitor being developed for the treatment of B cell malignancies and autoimmune diseases. In a KINOMEscan assay conducted in parallel against numerous kinases at a 1 μM drug concentration, Orelabrutinib was more selective than ibrutinib [74]. 2020 In this study, BTK was the only kinase targeted by orelabrutinib (with >90% inhibition). It is currently being investigated in clinical trials for lymphoid malignancies and autoimmune disorders [75,76]. In December 2020, orelabrutinib received initial approval in China for the treatment of patients with MCL and CLL/SLL who have received at least one prior treatment [76]. 

#### 3.4.3. Tirabrutinib

Tirabrutinib ((Velexbru^®^, ONO/GS-4059, Ono Pharmaceutical, Gilead Sciences) is another very potent and specific BTKi targeting BTK C481. It demonstrates greater selectivity than ibrutinib. The drug demonstrated potent activity in patients with CLL/SLL. Walter et al. report the results of an initial phase 1 study involving 90 RR patients with various B-cell malignancies [77]. In the CLL group, 96% (24/25) of patients achieved objective responses within the first three months of therapy. Elsewhere, a Japanese study examined its safety, efficacy, pharmacokinetics, pharmacodynamics, and prognostic biomarkers in patients with R/R primary central nervous system lymphoma (PCNSL) [78]. The most common AE was found to be mild diarrhea, occurring in 18% of the cases. In the CLL cohort, 14.3% of patients experienced drug-related grade 3 or 4 AE, with the most common being hematological toxicity. Further studies of tirarutinib with other targeted drugs are ongoing [79]. 

#### 3.4.4. SHR1459 

SHR1459 (TG 1701, EBI-1459; Reistone Biopharma, Jiangsu Hengrui Medicine Co., Lianyungang, China) is a second-generation, covalently bound, and irreversible second-generation BTKi currently under clinical development. This agent has been found to demonstrate superior selectivity to BTK compared to ibrutinib in in vitro kinase screening [80]. SHR1459 therapy, alone or in combination with ublituximab and umbralisib, is currently under clinical development in phase 1 trial in patients with R/R mature B cell neoplasms or CLL (NCT03671590; NCT04806035) [81]. 

#### 3.4.5. DTRMWXHS-12

DTRMWXHS-12 (DTRM-12) (NCT02900716) is a pyrazolo-pyrimidine derivative irreversible BTK inhibitor currently under phase 1 and phase 2 clinical trials for CLL and NHL [82]. DTRMWXHS-12, used alone and in combination with everolimus and pomalidomide, has yielded encouraging findings in several high-risk, multirefractory CLL and NHL patients, including those previously treated with ibrutinib, in simultaneous phase I studies [83]. DTRM-12 monotherapy was well tolerated across B cell malignancies and CLL in both studies. No dose-limiting toxicity (DLT) was observed, and MTD was not identified. PK studies demonstrate adequate target drug exposures at all dose levels. A phase II expansion cohort study of DTRMWXHS-12 in combination with everolimus and pomalidomide in patients with refractory or relapsed CLL and NHL is ongoing (NCT04305444). 

## 4. Reversible BTK Inhibitors

Reversible BTKi, such as pirtobrutinib and vecabrutinib, bind BTK non-covalently and do not require C481 to be present [84], thus overcoming this resistance mechanism. They can therefore inhibit BTK in the presence of the C481S mutation, and non-selective reversible BTK-i, including MK1026, may also overcome mutations within PLCG2 [85]. Results from ongoing studies of alternative BTK inhibitors will help define their role in CLL therapy as single drugs or in combination.

### 4.1. Pirtobrutinib

Pirtobrutinib (LOXO-305, Loxo Oncology/Lilly) is a highly selective, next generation BTKI that blocks the ATP binding site of BTK by noncovalent, non-C481-dependent binding, thus overcoming acquired resistance to covalent BTKis [86]. Recently, the results of the BRUIN phase 1/2 study of pirtobrutinib performed in mature B-cell lymphoid malignancies has been reported, with promising results [87]. The study enrolled 323 patients, including 170 with heavily pretreated R/R CLL. Among them, 25% of patients had 17p deletion, 30% *TP53* mutation, 19% 11q deletion, and 88% unmutated *IGHV*. The median number of previous lines of therapy was three (range 2–5). Moreover, 86% of patients had previously been treated with a BTKi, and 34% with venetoclax. In the total population of CLL patients, ORR was 63% including 79% in del(17p) and/or *TP53* mutated patients. In patients previously treated with other BTK inhibitors, ORR was 62%. Moreover, ORR was similar in patients who previously discontinued another BTKi for progression (67%) or toxicity (52%). Importantly, in patients with a BTK C481 mutation, ORR was 75%. The reported most common adverse events included fatigue (20%), diarrhea (17%), contusion (13%), and grade 3 or higher neutropenia (10%). Importantly, grade 3 atrial fibrillation/flutter were not observed, and only one patient with major bleeding was reported, caused by mechanical trauma. Only 1% discontinued treatment due to adverse events. Currently, pirtobrutinib is being compared to investigator’s choice in a phase 3 global, randomized, open-label study in CLL/SLL patients treated with at least a covalent BTK inhibitor. The choice consists of either idelalisib + rituximab or bendamustine + rituximab. (BRUIN CLL-321, NCT04666038). 

### 4.2. Vecabrutinib

Vecabrutinib (SNS-062, SNSS) is a selective, reversible, non-covalent, nanomolar potency and mutant BTKi [88]. In vitro studies have found it to demonstrate antitumor activity, even in cells that carry BTK Cys481Ser mutation. This dual activity is believed to result from vecabrutinib binding to the C481-independent BTK domain and suggests that it may have therapeutic potential in both covalent BTK relapse inhibitor disease and treatment-naive patients. Due to its very promising preclinical profile, this drug is being evaluated in 1b/2 phase study in patients with various B cell malignancies including CLL (NCT03037645) to potentially overcome ibrutinib resistance [89].

### 4.3. Fenebrutinib

Fenebrutinib (GDC-0853, Roche/Chugai Pharmaceutical) is another selective, reversible, non-covalent inhibitor of BTK that does not require interaction with the Cys481 residue for its activity [90]. Structural analyses have identified unique fenebrutinib-BTK interactions, which may explain its selectivity. The in vitro activity of the compound was also preserved towards BTK demonstrating single and double lesions of C481S, T474A, and T474S/C481S, respectively [91]. In addition to autoimmune disorders such as systemic lupus erythematosus and rheumatoid arthritis [92], studies are currently examining the therapeutic potential of fenebrutinib in 24 R/R patients with B-cell malignancies, 14 of whom with CLL. Of the latter, six had been previously treated with BTK and were C481S-positive [93]. The findings indicate an ORR of around 30% with only one CR. The most common grade 3 or higher AE was anemia (12.5%); however, fatigue (38%), nausea (33%), diarrhea (29%), thrombocytopenia (25%), and headache (21%) were also reported. Overall, 7 of the 14 patients with CLL achieved OR, with a mean duration of 2.5 months. Three fatalities were reported: one from pneumonia, another from H1N1 infection, and another due to disease progression.

### 4.4. Nemtabrutinib 

Nemtabrutinib (MK1026, *ARQ 531;* ArQule, Inc./Merck Sharp and Dohme) is able to reversibly inhibit of both wild-type and ibrutinib-resistant C481S-mutant BTK. It has a distinct selectivity profile with regard to kinases and has been found to demonstrate strong inhibition against a number of known key oncogenic factors, including those from the TEC, Trk, and Src kinase families. In two murine engraftment models of CLL (Eμ-TCL1 and Eμ-MYC/TCL1) resembling Richter transformation, nemtabrutinib was found to bestow increased survival compared to ibrutinib [94,95]. Nemtabrutinib was also observed to inhibit CLL cell survival and BCR-mediated activation of C481S BTK and PLCγ2 mutants; these are known to foster clinical resistance to ibrutinib. It also demonstrated greater efficacy in prolonging survival in animal models than ibrutinib and displayed in vitro activity in CLL cells with the ibrutinib resistance mutations BTKC481S or PLCγ2 [95]. Preliminary findings from a phase 1 escalation trial suggest that nemtabrutinib also has clinical activity against R/R B-cell lymphoid malignancies [95]. Overall, 14 out of the 47 patients treated with nemtabrutinib had PRs and additional 10 patients had stable disease. Responses were observed in patients with CLL, Richter’s transformation, diffuse large B-cell lymphoma (DLBCL), and follicular lymphoma (FL).

## 5. Resistance to BTK Inhibitors 

Acquired resistance to covalent inhibitors is observed in approximately 60% of long-term treated patients with CLL [96,97]. In most cases, BTK resistance is caused by the development of clones with mutated cysteine (C481) in the ibrutinib binding site [98,99]. Among CLL/SLL patients who progressed after ibrutinib administration, mutations were found in the BTK Cys481, SH2 (BTK Thr316), and BTK Thr474 binding domains of BTK. Cysteine-to-serine mutations at the C481 site allow further signaling, including activation of PLCγ2 and CARD11, thus bypassing inactive BTK and promoting the activation of distal BCR signaling. These changes result in cancer cell proliferation and migration. 

In CLL, the mechanism of acalabrutinib resistance is similar to that of ibrutinib and is related to BTK mutation [100]. In a study of 103 patients with CLL treated with acalabrutinib and routinely screened for BTK mutation, 22 were found to develop mutations. The median time from acalabrutinib initiation to mutation detection was 31.6 months. Among 16 patients with CLL progression, 11 (69%) demonstrated BTK C481 mutations, including C481S in 10 patients, C481R in one and C481Y in another. Four patients demonstrated additional mutations to BTK C481S, including BTK T474I in one, BTK C481R in another, and PLCG2 in two others. All these mutations had previously been identified in patients with ibrutinib resistance. 

Several strategies for overcoming resistance to BTK have been investigated [97]. In particular, the third-generation, reversible, noncovalent BTK inhibitors display inhibitory activities against both BTK and BTKC481 mutants and have the potential to overcome resistance to covalent inhibitors caused by BTKC481 mutation [101]. Noncovalent binding BTK inhibitors such as pirtobrutinib, vecabrutinib, and MK1026 continue to inhibit BTK in the presence of the C481S mutation and have been found to be effective against C481 mutants [102]. Vecabrutinib and pirtobrutinib are more specific and inhibit wild-type and C481S-mutated BTK. Nemtabrutinib inhibits additional targets and has demonstrated activity in the presence of mutated PLCG2 [103]. Sequencing and combination therapies can also overcome BTKi resistance. Venetoclax is an active drug in CLL patients relapsed after ibrutinib [104]. PI3K inhibitors, such as idelalisib, duvelisib, or umbralisib, have also demonstrated therapeutic activity in CLL patients, previously treated with BTK inhibitors, who developed progression on treatment [105]. CD3×CD19 bispecific antibodies can mediate effective killing of CLL cells regardless of IGVH and TP53 mutational status, irrespective of sensitivity to ibrutinib. Recent studies have also found CD3×CD19 bispecific antibodies to be effective in killing ibrutinib-resistant CLL cells [106]. Chimeric antigen receptor-modified T (CAR-T) cell therapy has been investigated in CLL patients refractory to ibrutinib and showed promising activity. In the TRANSCEND CLL 004 phase 1/2 study, performed in R/R CLL patients failing on previous ibrutinib therapy, patients received CD19-directed CAR-T therapy; one-half of the patients had previously failed on both ibrutinib and venetoclax [107,108]. In addition, another new strategy for overcoming BTKi resistance is proteolysis-targeting chimera (PROTAC)-induced degradation of BTK. PROTAC has been found to be effective in a mouse model of the BTK C481S mutation [109].

## 6. Adverse Events

BTK inhibitors have unique toxicities that require monitoring. As such, it is essential to provide optimal management to achieve the best possible outcomes for patients [110,111]. The most common reason for discontinuing ibrutinib is toxicity, particularly the AEs specific for this group of drugs, such as atrial fibrillation (AF), bleeding events, arthralgias, rash, diarrhea, and cytopenias [112]. Ibrutinib discontinuation caused by AEs, mainly AF, arthralgias, rash, diarrhea, and bleeding events, was observed in 4–26%. Acalabrutinib and zanubrutinib are selective next-generation covalent BTK inhibitors, with less off-target activity than ibrutinib and better tolerability. The most common AE associated with acalabrutinib with acalabrutinib is headache, observed in 22–51% of patients. The incidence of AF is lower than for ibtrutinib. In patients treated with zanubrutinib, the most common grade ≥3 AEs were neutropenia and infections. The following section summarizes the key BTK inhibitor-related AEs in patients with CLL and strategies for their management.

### 6.1. Bleeding and Bruising

Bleeding and bruising are frequently observed AEs in patients treated with BTKis [113]. In a systematic review and pooled analysis of four randomized controlled trials, ibrutinib treatment was associated with an increase in all-grade bleeding (4.85% vs. 1.55%, RR = 2.93, *p* = 0.03) compared to control treatments. This is believed to be partly due to the off-target TEC kinase inhibition and platelet inhibitory mechanisms [114,115]. Acalabrutinib did not induce the platelet dysfunction or inhibition of platelet aggregation observed with ibrutinib [116]. However, bleeding complications were also observed in patients treated with acalabrutinib. In a pooled safety analysis of zanubrutinib monotherapy in patients with B-cell malignancies, bruising occurred in 25% of patients and major hemorrhage in 4% [117]. The risk of bleeding and bruising is more common in patients simultaneously treated with antiplatelet drugs and anticoagulants; in these patients, serious life-threatening bleedings were observed. The ibrutinib-associated risk of bleeding can decrease by prohibiting the use of oral anticoagulants and by avoiding CYP3A4 drug–drug interactions [118]. It is therefore recommended to stop BTKi treatment for three days before and after any minor invasive procedure, and for seven days before and after a major surgical procedure, to decrease the risk of bleeding. Patients considered to receive BTKis should be advised to stop aspirin treatment or reduce the dose to 81 mg if necessary. However, patients with episodes of bleeding or bruising should not receive anticoagulation treatment if possible. Achieving optimal anticoagulation therapy in patients with atrial fibrillation during BTKi treatment is a difficult challenge. For patients with new AF, anticoagulation should be recommended to decrease the risk of stroke. Apixaban is usually suggested following an analysis of the risks and benefits. However, the need for BTKi continuation should be considered. 

### 6.2. Cardiovascular Complications

Cardiotoxicity is an important complication of treatment with ibrutinib and other BTKis. A pooled analysis of four randomized controlled clinical trials found ibrutinib to be associated with an increased risk of AF and flutter compared to other treatments (3.03% vs. 0.80%, RR = 3.80, *p* = 0.003 [119]. In another study of 178 patients with CLL, those who discontinued ibrutinib-based therapy gave atrial fibrillation as the most common reason for discontinuation [120]. Long-term data indicate that approximately 20% of ibrutinib-treated patients develop incidences of arrhythmia [121,122]. 

A review of the Food and Drug Administration Adverse Event Reporting System (FAERS) database by Grewel et al. examined the cardiovascular complications associated with novel agents in CLL, including ibrutinib and acalabrutinib [123]. A total of 6074 cardiac adverse events were identified. Of the examined agents, ibrutinib was found to have the highest risk of cardiac adverse events (4832/36581; 13.2%). The ibrutinib group also demonstrated a higher frequency of AF (41.5%) than in the group treated with acalabrutinib (age-adjusted OR = 2.21, 95% CI = 1.25–3.90, *p* = 0.005). A pooled safety analysis of zanubrutinib monotherapy in a group of 779 patients with B-cell malignancies found AF to occur in 3%; for such patients, multidisciplinary care is indicated to optimize anticoagulant treatment and cardiac management. Ibrutinib is associated with a significantly increased risk of hypertension [124]. In an early-phase study of ibrutinib in CLL, 23% of patients developed new or worsened existed hypertension, and follow-up data suggest a continual increase in the incidence of hypertension over time [125,126]. In a recent analysis by Dickerson et al., 78.3% of patients treated with ibrutinib developed new or worsened hypertension over a median observation period of 30 months [127]. Hypertension developed in 71.6% of patients, with a time to 50% cumulative incidence of 4.2 months. In addition, 17.7% of the patients developed high-grade hypertension with blood pressure above 160/100 mm Hg. Taken together, these findings suggest that ibrutinib treatment results in a higher risk of hypertension and greater severity, as well as an increased risk of cardiotoxic events; however, acalabrutinib treatment is associated with a lower burden of hypertension than ibrutinib. Grade ≥3 hypertension was also observed in 5% of patients with lymphoid malignancies treated with zanubrutinib [127]. 

### 6.3. Cytopenias

Cytopenias are relatively frequent AEs in patients treated with BTKi, but they are usually not serious and are typically managed with supportive care and/or treatment interruption. Severe cytopenias are rarer in CLL patients treated with BTKis than other antileukemic drugs and usually do not warrant dose reduction. Among patients treated with zanubrutinib, grade ≥3 neutropenia was observed in 23%, thrombocytopenia in 8%, and anemia in 8% [128]. BTK inhibitor treatment should be withheld from patients demonstrating grade ≥3 neutropenia and infection or fever until resolution to baseline or grade 1; following this, the treatment can be restarted at a reduced dose. In addition, patients demonstrating grade 3 thrombocytopenia with bleeding or grade 4 thrombocytopenia should not receive BTK inhibitor treatment until resolution to baseline or grade 1, following which, BTKi treatment can be restarted at a reduced dose. If cytopenia occurs for a fourth time, the BTKis should be discontinued. Treatment-emergent autoimmune cytopenias were observed in 1% of patients during ibrutinib therapy [129]. In a phase 2 study evaluating acalabrutinib monotherapy in 134 R/R CLL patients, only one case of autoimmune hemolytic anemia (AIHA) recurrence was noted among 11 patients with a history of autoimmune cytopenia. In addition, ibrutinib induces rapid and durable responses when used to treat AIHA developed in CLL patients [129].

### 6.4. Infections

BTK plays an important role in immunity, participating in numerous pathways including B cells, T cells, and macrophages. It is therefore a driving factor in both lymphoproliferative disorders and response to infection [130,131]. BTKis are considered less immunosuppressive and safer than other chemotherapeutic drugs and have been proposed as useful agents for reconstituting humoral immunity and protecting against infection in patients with CLL [132]. In addition, ibrutinib can inhibit inflammation induced by bacterial infection. 

However, several clinical trials indicate an increased risk of infection, including upper respiratory tract infection, pneumonia, cellulitis, and sepsis [126,133,134,135]. In a recent retrospective, single-center study of patients treated with ibrutinib, 11.4% of the patients developed serious infections requiring hospitalization or parenteral therapy, including 4.2% with invasive fungal infections [133]. In another study, the incidence of one or more serious infections in patients with hematologic malignancies receiving ibrutinib therapy was 18.0%; most (16.1%) were bacterial in nature and 4.3% multiple infections [134]. In a pooled analysis from 1476 ibrutinib-exposed patients, the incidence of Grade ≥3 infection was 21%. Grade ≥3 infection was observed in 27% patients receiving zanubrutinib. This value is similar to that reported in a pooled analysis from ibrutinib-exposed patients. 

Mauro et al. analyzed the risk factors of infections in 494 CLL patients treated with ibrutinib [136]**.** They found pneumonia, grade 3 or higher non-opportunistic infections, and opportunistic infections in 32% of patients, with an overall incidence rate per 100 person-year of 15.3%. Infections also caused the permanent discontinuation of ibrutinib in 9% of patients. Severe infection in the year before starting ibrutinib, chronic obstructive pulmonary disease, and two or more prior treatments were associated with a two to threefold increase in the rate of infections. 

At present, the individual contribution of BTKis to the risk of serious infection is unclear. A body of evidence indicates that this risk is influenced by various additional factors such as concurrent steroid use, neutropenia, and prior chemo- or immunochemotherapy. In patients with multiple risk factors, targeted antimicrobial prophylaxis may reduce the risk of infection associated with BTKi use. Vaccinations (e.g., against influenza, COVID-19, and pneumococcus) are recommended before treatment initiation [137,138]. Recombinant, adjuvanted varicella-zoster virus vaccine should be also considered [139]. Intravenous immunoglobulin supplementation is also useful for patients with recurrent infections and hypogammaglobulinemia.

### 6.5. Arthralgias and Myalgias

The occurrence of arthralgias/myalgias is a common AE in CLL patients treated with ibrutinib in both upfront and R/R settings, with an increased risk observed at longer treatment durations [139,140]. In clinical trials and retrospective studies, arthralgias and myalgias were noted in 11–36% of patients [139,140,141,142,143,144]. In a recent analysis, 76 of 214 (36%) patients with CLL treated with ibrutinib, either as a single agent or in combination, developed arthralgias/myalgias during follow-up, with a median follow-up of 34.5 months [140]. Most patients (79%) had grade 1 or 2 toxicity, and 28% continued ibrutinib with resolution of symptoms. More effective toxicity management was observed for dose holds of ibrutinib than dose reduction. However, 63% of patients with grade 3 or higher toxicity discontinued ibrutinib treatment, indicating that this subgroup does not tolerate ibrutinib. For some patients, the use of non-steroidal anti-inflammatory drugs, acetaminophen, or corticosteroids can temporarily reduce symptoms; however, these drugs may exacerbate the risk of bleeding and should be used with caution [141]. The mechanisms behind of ibrutinib-induced arthralgias/myalgias remain unclear. 

Additional studies are needed to determine the mechanism of ibrutinib-related arthralgias/myalgias and develop optimal management strategies. Rhodes et al. recommend continuing ibrutinib at the current dose in the case of grade 1 or 2 arthralgias/myalgias, as long as the symptoms do not interfere with activities of daily living, as most symptoms can resolve spontaneously [139]. If symptoms affect daily activities, dose reduction should be advised. If there is no improvement at a lower dose, further dose reduction or a dose hold until improvement in symptoms should be recommended. However, if symptoms recur with re-challenge after a dose hold, ibrutinib should be permanently discontinued, and the use of alternative CLL-directed therapies is recommended. In cases of grade 3 or higher arthralgias or myalgias, the dose hold should be maintained until resolution of symptoms. If the symptoms resolve, re-challenging can be performed with a lower dose. If the symptoms do not recur, it is advisable to continue with ibrutinib at a reduced dose, rather than attempting to escalate. If symptoms recur, ibrutinib should be discontinued, as well as other CLL-directed therapies, and BTKi considered. Replacing ibrutinib with acalabrutinib is a reasonable option, as acalabrutinib seems not cause myalgias. A recent study indicates that approximately two-thirds of patients with ibrutinib-induced arthralgias/myalgias did not experience recurrent symptoms following acalabrutinib treatment [144].

### 6.6. Dermatologic Complications

Approximately 20% of patients treated with ibrutinib or acalabrutinib demonstrate rashes, which have been associated with EGFR inhibition and infiltration of inflammatory cells [145,146]. Witholding the BTKi is usually recommended and if the rash resolves, the BTK inhibitor can be resumed. However, if the rash recurs, a dose reduction is indicated. In addition, erythema nodosum was observed in patients treated with ibrutinib. Both skin symptoms usually respond to corticosteroids or dose holds. Textural changes can also be observed in the hair and nails, with long-term ibrutinib therapy being associated with a higher incidence of brittle fingernails or toenails [147]. However, these manifestations tend to have a gradual onset, typically around nine months, and are not dose-limiting toxicities. Recommended treatments are biotin supplementation and nail oil application.

### 6.7. Headaches

Headaches are noted approximately in 40% of patients treated with acalabrutinib and are rarely observed in patients receiving ibrutinib [148]. Usually, headaches occur early after treatment initiation and the incidence decreases over time. This AE is usually a manageable toxicity and does not influence the continuation of BTKi treatment.

### 6.8. Diarrhea

Diarrhea is commonly observed AE in patients treated with BTKi. It is observed mainly in the first six months from beginning therapy, with a frequently observed self-limited course [148,149]. Similar incidence is observed between patients treated with ibrutinib and acalabrutinib [148,149]. Temporary drug holds should be considered in the case of grade ≥3 diarrhea.

## 7. BTK Inhibitors and the COVID-19 Pandemic

As BTK regulates the activity of macrophages, it has been suggested that BTK inhibitors may have a therapeutic role in COVID-19 patients. This hypothesis was supported by the observation that activated macrophages were responsible for the hyperinflammatory immune response observed in patients with severe COVID-19 symptoms [150,151]. In patients with severe COVID-19, treatment with acalabrutinib was associated with rapid clinical improvement, with increased oxygenation and reduced inflammation being observed in the majority of patients. However, it is important to note that BTK inhibitors impair the innate immunity and increase susceptibility to infections. Despite this, long-term BTKi therapy may improve recovery of humoral immune function, decrease infection rates, and protect CLL patients from lung injury in the event of COVID-19 infection [152]. The use of BTKi therapy is supported by a pilot study in six CLL patients with COVID-19 infection who continued therapy [153].

## 8. Future Directions

The BTK inhibitors ibrutinib and acalabrutinib, and venetoclax, induce long-lasting remissions in most patients with CLL, with or without CD20 antibodies. However, their use may eventually be associated with the development of clinical resistance or unacceptable toxicity. When used as single drugs in CLL, BTK inhibitors are given continuously until disease progression or unacceptable toxicity. BTKi combinations are given for a limited time [154]. In addition, BTKi resistance can be overcome and toxicity can be reduced with the use of combination therapies. Several ongoing studies are therefore focusing on time-limited combination therapy strategies with incorporate venetoclax and CD20 monoclonal antibodies. Such combination therapy may also elicit a deeper response. However, the toxicity, financial costs, and efficacy of the drug should be taken into account when deciding on the optimal therapeutic solution.

Hope in the treatment of CLL resistant for currently available drugs is offered by combining BTKi with modern antibody-based therapies [155]. These include bispecific antibodies, antibody–drug conjugates, antibody-associated immune modulation, and other agents. Bispecific antibodies have shown antileukemic activity against CLL cells in vitro and in a xenograft mouse model. The bispecific anti-CD3 and anti-CD20 antibody epcoritamab is currently under evaluation in a phase 1 trial for R/R CLL [NCT04623541]. In response to clinical and pharmacological considerations, ibrutinib therapy has been tested in combination with the proteasome inhibitor carfilzomib and the HDAC inhibitor abexinostat in preclinical models, and initial clinical trials for such ibrutinib-based combination therapies are ongoing [156,157]. 

Another option is the introduction of novel, more specific, and less toxic BTKi agents that can be more effective and safer than ibrutinib and second generation BTKis. The next generation of reversible BTKis bind non-covalently to BTK but are still active in CLL cells harboring the most common resistance mutations (BTK C481S). Two third-generation reversible BTKis, pirtobrutinib and nemtabrutinib, yielded promising results in early clinical trials and are now under examination in phase 2 and phase 3 trials (NCT05023980, NCT04965493, NCT04728893, NCT03162536).

Recently, other genetic alterations have been identified in CLL that may play a role in the prognosis and evolution of CLL, including those associated with TLR, MAPK, and Notch signaling pathways [158]. However, no information is currently available regarding the efficacy of novel targeted drugs, including BTK inhibitors, in patients with recurrent mutations other than *TP53* and *NOTCH1*. Alternatively, another target for potentially useful novel agents in CLL is the Toll-like receptor (TLR) intracellular signaling pathway, which is open to modification by interleukin-1 receptor-associated kinase (IRAK) inhibitors, monoclonal antibodies, oligonucleotides, lipid-A analogs, and microRNA.

Ibrutinib and IRAK4 inhibitors demonstrated better antitumor activity against CLL cells in vitro when used in combination than when each agent was used alone [159,160]**.** The IRAK4 inhibitor CA-4948 is currently under investigation in R/R CLL and other indolent B-cell malignancies as a single agent or in combination with ibrutinib (NCT03328078). Other possible targets for CLL drugs have also been identified recently. For example, the ERK inhibitor ulixertinib decreased ERK phosphorylation in MAPK-mutated CLL cells. The MEK1/2 inhibitor binimetinib also inhibited CLL survival induced by stroma-conditioned media and phorbol myristylate (PMA) when used alone or in combination with venetoclax [161]. The combination of these agents with BTKi may be an effective treatment strategy for CLL. Finally, the SYK inhibitor entospletinib also appears to be a promising drug with clinical activity for patients previously treated with BCR-pathway inhibitors; treatment was found to yield OS of 32.7% and PFS of 5.6 months when used in monotherapy in patients with R/R CLL who had been previously treated with ibrutinib, spebrutinib, idelalisib, or umbralisib [162].

Treatment options in patients resistant to novel agents like BTKi, PI3k inhibitors, and venetoclax are limited [163]**.** Cellular treatment options such as allogeneic hematopoietic stem cell transplantation (allo HCT), chimeric antigen receptor (CAR) T-cells, and CAR NK-cells appear promising, especially in patients who failed BTKi combinations [164]. BTK inhibitors combined with CART cell treatment has been also tested in patients with CLL, showing marked improvement in CART cell activity in patients with CLL [165]. The development of more specific targeted therapies and novel combination treatments will help to design personalized effective treatments and strategies to improve the outcome of patients with CLL

## 9. Conclusions

The approval of BTKi, PI3K inhibitors, and Bcl-2 inhibitors has improved the treatment efficacy of patients with CLL. These drugs are highly effective in previously untreated and R/R CLL, including high-risk patients with 17p deletion and/or TP53 mutation. However, the first-in-class BTKi ibrutinib has some limitations, such as the development of resistance, mainly due to C481 BTK mutations, and its toxicity, including atrial fibrillation and bleeding, mainly related to off-target activity. Second-generation irreversible BTKis, including acalabrutinib and zanubrutinib, show greater selectivity than ibrutinib, with minimal off-target activity and better safety profiles. New, reversible BTKis such as fenebrutinib, pirtobrutinib, vecabrutinib, and nemtabrutinib inhibit BTK in the presence of the C481S mutation and are effective in patients resistant to irreversible BTK inhibitors. Robust results have been reported for the combination of BTK inhibitors with venetoclax and/or with CD20 antibodies. These combinations induce deeper responses and can be used for limited periods. However, choosing optimal BTK combinations and proper patient selection needs further investigation.

## Figures and Tables

**Figure 1 cancers-14-00771-f001:**
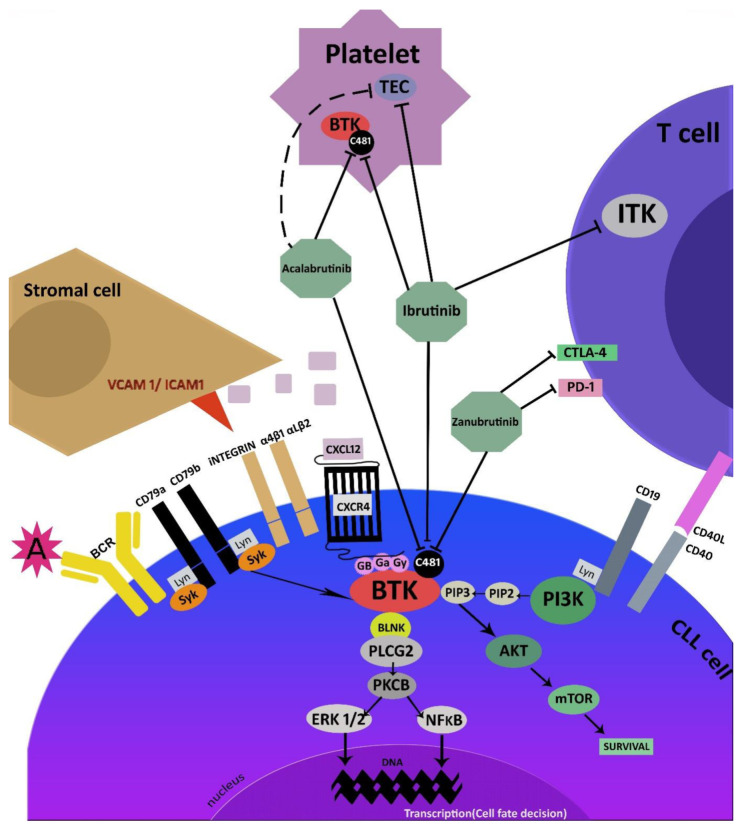
Signaling pathways involved in the mechanisms of action of Bruton kinase inhibitors in chronic lymphocytic leukemia (CLL) cells. Abbreviations: AKT—protein kinase B, BCR—B-cell receptor, BLNK—B-cell linker protein Btk: Bruton’s tyrosine kinase, CTLA-4: cytotoxic T lymphocyte-associated antigen-4, C481—cysteine residue, CXCL12—C-X-C motif chemokine ligand 12, CXCR4—C-X-C chemokine receptor type 4, EGFR: epidermal growth factor receptor, ERK1/2—extracellular signal-regulated kinases 1 and 2, Gα, G β, Gϓ: G protein subunits, ITK—IL2-inducible T-cell kinase, Lyn—member of the Src kinase family, NFkB—nuclear factor kappa B, PI3K—phosphoinositide 3-kinase, PLCG2—phospholipase gamma 2, PKCB—protein kinase C beta, PKCB: protein kinase C beta, PD-1—programmed death-ligand 1, PIP1, PIP2—phosphatidylinositols 1 and 2, Syk—spleen tyrosine kinase, TEC—tyrosine kinase expressed in hepatocellular carcinoma.

**Table 1 cancers-14-00771-t001:** Irreversible and reversible Brutton tyrosine kinase inhibitors approved or in clinical trials in chronic lymphocytic leukemia.

BTKi	Binding	T1/2 [hours]	IC50 [nM]	Dosing	Clinical Trials in CLL
Ibrutinib (PCYC-1102)	Covalent irreversible C481	4–8	0.5	420 mg	NCT04771507 NCT03513562 NCT02912754
Acalabrutinib (ACP-196)	Covalent irreversible C481	0.9	5.1	100 mg twice a day	NCT04008706 NCT04930536 NCT04722172
Zanubrutinib (BGB-3111)	Covalent irreversible C481	2–4	0.5	160 or 320 mg twice a day	NCT04116437 NCT04458610 NCT03824483 NCT04282018 NCT04515238 NCT03336333
Spebrutinib (CC-292)	Covalent irreversible C481	8–24	<0.5	1000 mg	NCT02031419
Tirabrutinib (ONO/GS-4059)	Covalent irreversible C481	NA	5.6	80 mg	NCT03740529 NCT03162536
Orelabrutinib (ICP-022)	Covalent irreversible C481	~1.5–4 h	1.6	150 mg	NCT03493217 NCT04014205
SHR1459 (TG-1701)	Covalent irreversible C481	NA	3	300 mg	NCT03671590; NCT04806035
DTRMWXHS-12 (DTRM-12)	Covalent irreversible C481	~4	NA	200 mg	NCT02900716 NCT04305444
Pirtobrutinib (LOXO-305)	Non-covalent reversible	NA	0.85	200 mg	NCT05023980 NCT04965493 NCT05024045 NCT04666038
Vecabrutinib (SNS-062)	Non-covalent reversible	6.6-8	24	25 mg escalated to 500 mg	NCT03037645
Fenebrutinib (GDC-0853)	Non-covalent reversible	2.2	0.91	200 mg twice a day	NCT01991184
Nemta brutinib (ARQ 531)	Non-covalent reversible	NA	0.85	65-100 mg	NCT04728893 NCT03162536

Abbreviations: BTKi—Bruton tyrosine kinase inhibitors, NA—not available.

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
