# Peer review of "The Role of Bruton’s Kinase Inhibitors in Chronic Lymphocytic Leukemia: Current Status and Future Directions"

_cancers, 2022, doi:10.3390/cancers14030771_

Round 1
Reviewer 1 Report
This review summarizes the pharmacology, clinical efficacy, safety, dosing, as well as drug interactions in CLL patients undergoing BTK inhibitors treatment. Future options for the use of these molecules are also taken into account.
Overall, this is an interesting and well organized review on data so far collated on the role of Bruton’s kinase inhibitors in CLL. There are several limitations that the authors should address in order for the paper to be more complete and publishable in a journal such as Cancers. The following concerns require to be approached to improve the appeal of the ms.
General comments
Limitations of this review are mostly related to the lack of discussions and/or interpretations of the collated data, or suggestions on how the next steps of research on these molecules could be. In addition, cutting edge topics currently on debate on the matter are not properly addressed.
The actual challenge is to develop therapeutic strategies that achieve deeper responses. The issue of MRD is therefore absolutely central at this time not only in terms of evaluation of the efficacy of different drugs or their combinations but also for setting the monitoring of these patients. It is surprising that the issue of MRD is taken into account only marginally (when reporting the activity of Zanobrutinib). No mention has been made made about the MRD induced by Ibrutinib (e.g., CLARITY and CAPTIVATE studies) and by Acalabrutinib (e.g., ASCEND). This topic is nowdays subject of many debates (see Leukemia 35:3364,2021) and deserves appropriate relevance.
Always in terms of achieving deep responses, it would be useful for the reader to know the effects of these new drugs on the altered pathways that contribute to the development and progression of CLL, including TP53 and other recurrent pathways (see Cancers 13: 3150, 2021).
Another general issue that is very current right now rests on the ongoing trials investigating triplet vs doublet combination therapies. Since the aim of this review also deals with prospecting “future directions”, mention of the current clinical trials is mandatory. In particular, the underlying issues, i.e. the balance between efficacy versus safety, feasibility and sustainability, of these combination therapies would have expected to be properly discussed. In the present form of the ms the above topics are only marginally mentioned in the Conclusions section. I suggest that the issue be taken into account in a separate chapter and be adequately expanded.
Again in terms of efficacy of BTK inhibitors and patient’s monitoring, and having in mind the “future directions”, it would be nice to discuss the model of temporal dinamics upon Ibrutinib treatment in CLL made by single-cell immune profiling (Nat Commun. 11:577, 2020) as an innovative frontier for the management of these patients.
Specific comments
A figure depicting the kinome profile of the up-to-date approved BTK inhibitors would be helpful to the reader to immediately get the message of potency and selectivity of different compounds. As you know, figures provide added value to the piece, especially in review articles.
It has been extensively demonstrated that in CLL the Src kinase Lyn plays the pivotal role in the pathogenesis of the diseases, mainly by its constitutive phosphorylation. Please correct/complete the sentence on page 3 lines 86-88.
Tables 2 and 3 summarize the most important clinical trials. However, to better compare the studies, I suggest to add i) the median follow-up of each study (which is pretty different among studies), ii) Rab and OS as for example 3 years PFS, iii) discontinuation rates.
Why did the authors decide to exclude reporting studies with ibrutinib + venetoclax? Authors should explain this choice, since this fixed duration combination will become relevant in next future. Studies with ibrutinib plus ublituximab are also missing.
Page 10, line 406. Data from the SEQUOIA trial arm D (Zanobrutinib plus Venetoclax) have recently been presented at the last ASH meeting.
Why did the Authors take Evobrutinib, Elsobrutinib, Tolebrutinib, Rilzabrutinib into account ? They are under investigation for immune-mediated diseases and data on CLL are not available. I suggest to remove them, thus getting room to face the problems reported above.
In the paragraph dealing with cardiovascular events and infections, I suggest to summarize the most important ibrutinib-induced atrial fibrillation scores as well as the issue of ibrutinib-induced infections (see Cancers 13:3240,2021).
Minor Points
- Please replace IgVH with IGHV gene.
- Page 7 line 236. I think that the reference Table is #1 rather than Table #2.
Author Response
Reviewer 1.
This review summarizes the pharmacology, clinical efficacy, safety, dosing, as well as drug interactions in CLL patients undergoing BTK inhibitors treatment. Future options for the use of these molecules are also taken into account.
Overall, this is an interesting and well organized review on data so far collated on the role of Bruton’s kinase inhibitors in CLL. There are several limitations that the authors should address in order for the paper to be more complete and publishable in a journal such as Cancers. The following concerns require to be approached to improve the appeal of the ms.
General comments
Limitations of this review are mostly related to the lack of discussions and/or interpretations of the collated data, or suggestions on how the next steps of research on these molecules could be. In addition, cutting edge topics currently on debate on the matter are not properly addressed.
Response: We thank the Reviewer for positive review of our paper . We have added the discussions and interpretations of the data whenever possible.
The actual challenge is to develop therapeutic strategies that achieve deeper responses. The issue of MRD is therefore absolutely central at this time not only in terms of evaluation of the efficacy of different drugs or their combinations but also for setting the monitoring of these patients. It is surprising that the issue of MRD is taken into account only marginally (when reporting the activity of Zanobrutinib). No mention has been made made about the MRD induced by Ibrutinib (e.g., CLARITY and CAPTIVATE studies) and by Acalabrutinib (e.g., ASCEND). This topic is nowdays subject of many debates (see Leukemia 35:3364,2021) and deserves appropriate relevance.
Response: The MRD considerations were added including the monitoring of the patients. MRD induced by Ibrutinib (e.g., CLARITY and CAPTIVATE studies) and by Acalabrutinib (e.g., ). are also discussed . The ref Leukemia 35:3364,2021) is also added.. Finally, fixed-duration treatment with ibrutinib and venetoclax was investigated in a phase 2 study in R/R patients (CLARITY), and in previously-untreated patients (CAPTIVATE) [46, 47]. In the CLARITY study, the primary end point was eradication of minimal residual disease (MRD) after 12 months of treatment with ibrutinib and venetoclax. Therapy was given for a limited period and then stopped if patients achieved deep remission with MRD negativity. Undetectable MRD was defined as fewer than one CLL cell in 10,000 leukocytes. Out of 53 evaluated patients, MRD negativity was achieved in 28 (53%) in the peripheral blood (PB) and in 19 (36%) in the bone marrow (BM) after 12 months, with an OR rate of 89% and CR 51%. After a median follow-up of 21.1 months, all patients were alive and only one patient demonstrated progression. This is the first study to demonstrate that a combination of two drugs targeting the key pathogenetic pathways in CLL induced MRD-negative responses in a significant number of patients.
In the CAPTIVATE study 159 older/unfit previously untreated patients with CLL/SLL received three cycles of ibrutinib (420 mg/d orally) and subsequently 12 cycles of ibrutinib and venetoclax. ORR was 96% and CR was achieved by 88 patients (55%), including 78 (89%) patients with CR duration one year or longer. Undetectable MRD was achieved in 77% of patients in PB and 60% of patients in BM. A maintained response after two years was observed in most patients (90%) compared with 41% for chlorambucil plus obinutuzumab. Two-year PFS was 95% and two-year OS was 98%. The most common grade 3/4 AEs were neutropenia (33%), hypertension (6%), and decreased neutrophil count (5%). Fixed-duration ibrutinib-venetoclax combination therapy induced deeper and better sustained responses than chlorambucil-obinutuzumab in previously-untreated older/unfit patients with CLL.
In the GLOW open-label, randomized phase 3 study, 106 previously-untreated patients were randomized to receive ibrutinib plus venetoclax and 105 to receive chlorambucil plus obinutuzumab (Table 3) [48]. Patients with del(17p) or TP53 mutations were excluded. The participants received 420 mg/day ibrutinib for three months, followed by 12 cycles of ibrutinib plus venetoclax. In the control arm, the patients received six cycles of chlorambucil plus obinutuzumab at standard doses. The independent reviev commitee (IRC) found ORR values of 86.6% for the ibrutinib-venetoclax arm and 84.8% for the chlorambucil-obinutuzumab arm. However, the ibrutinib-venetoclax arm yielded a significantly higher CR rate (38.7%) than chlorambucil-obinutuzumab (11.4%) (p<0.0001). The undetectable MRD in BM scores at three months following end of treatment were 51.9% and 17.1%, respectively (p<0.0001). With a median follow-up of 27.7 (range, 1.7 to 33.8) months, median IRC-assessed PFS was not reached for the ibrutinib-venetoclax arm and 21.0 months for the chlorambucil-obinutuzumab arm; estimated 24-month PFS rates were 84.4% versus 44.1%, respectively. In addition, 90% of responders to ibrutinub-venetoclax demonstrated maintained response after two years compared with 41% of responders to chlorambucil-obinutuzumab.
Time-limited treatment with acalabrutinib, venetoclax, and obinutuzumab was also investigated in patients with previously-untreated CLL in a phase 2 study [57]. Therapy consisted of acalabrutinib alone for cycle 1, and acalabrutinib with obinutuzumab for six cycles. Venetoclax was used from the beginning of cycle 4 until day 1 of cycle 16 or day 1 of cycle 25. Patients with CR and undetectable MRD in the BM could discontinue therapy at the start of cycle 16 or at the start of cycle 25 if they were at least in PR. Undetectable MRD was defined as less than 1 CLL cell per 10 000 leucocytes, as measured by four-colour flow cytometry. At cycle 16 day 1, 14 (38%) of 37 patients obtained CR with undetectable MRD in the BM. An ongoing phase 3 study (NCT03836261) is currently comparing the effects of combined acalabrutinib and venetoclax, with and without obinutuzumab, with the investigator's choice of chemoimmunotherapy in previously untreated CLL
Always in terms of achieving deep responses, it would be useful for the reader to know the effects of these new drugs on the altered pathways that contribute to the development and progression of CLL, including TP53 and other recurrent pathways (see Cancers 13: 3150, 2021).
Response: The topic is discussed according to the reviewer suggestion in the future direction part.
Another general issue that is very current right now rests on the ongoing trials investigating triplet vs doublet combination therapies. Since the aim of this review also deals with prospecting “future directions”, mention of the current clinical trials is mandatory. In particular, the underlying issues, i.e. the balance between efficacy versus safety, feasibility and sustainability, of these combination therapies would have expected to be properly discussed. In the present form of the ms the above topics are only marginally mentioned in the Conclusions section. I suggest that the issue be taken into account in a separate chapter and be adequately expanded.
Response: The trials investigating triplet and doublet combination therapies are added for ibrutinib, acalabrutinib and zanubrutinib. In addition a separate chapter 8 (Future directions is included and combination therapies discussed
- Future directions
The BTK inhibitors ibrutinib and acalabrutinib, and venetoclax, induce long-lasting remissions in most patients with CLL, with or without CD20 antibodies. However, their use may eventually be associated with the development of clinical resistance or unacceptable toxicity. When used as single drugs in CLL, BTK inhibitors are given continuously until disease progression or unacceptable toxicity. BTKi combinations is given for a limited time [154]. In addition, BTKi resistance can be overcome and toxicity can be reduced with the use of combination therapies. Several ongoing studies are therefore focusing on time-limited combination therapy strategies with incorporate venetoclax and CD20 monoclonal antibodies. Such combination therapy may also elicit a deeper response. However, the toxicity, financial costs and efficacy of the drug should be taken into account when deciding on the optimal therapeutic solution.
Hope in the treatment of CLL resistant for currently-available drugs is offered by combining BTKi with modern antibody-based therapies [155]. These include bispecific antibodies, antibody-drug conjugates, antibody-associated immune modulation and other agents. Bispecific antibodies have shown antileukemic activity against CLL cells in vitro and in a xenograft mouse model. The bispecific anti-CD3 and anti-CD20 antibody epcoritamab is currently under evaluatin in a phase 1 trial for R/R CLL [NCT04623541]. In response to clinical and pharmacological considerations, ibrutinib therapy has been tested in combination with the proteasome inhibitor carfilzomib and the HDAC inhibitor abexinostat in preclinical models, and initial clinical trials for such ibrutinib-based combination therapies are ongoing [156,157].
Another option is the introduction of novel, more specific and less toxic BTKi agents that can be more effective and safer than ibrutinib and second generation BTKis. The next generation of reversible BTKis bind non-covalently to BTK, but are still active in CLL cells harboring the most common resistance mutations (BTK C481S). Two third-generation reversible BTKis, pirtobrutinib and nemtabrutinib, yielded promising results in early clinical trials and are now under examination in phase 2 and phase 3 trials (NCT05023980, NCT04965493, NCT04728893, NCT03162536).
Recently, other genetic alterations have been identified in CLL that may play a role in the prognosis and evolution of CLL, including those associated with the TLR, MAPK and Notch signaling pathways [158]. However, no information is currently available regarding the efficacy of novel targeted drugs, including BTK inhibitors, in patients with recurrent mutations other than TP53 and NOTCH1. Alternatively, another target for potentially useful novel agents in CLL is the toll-like receptor (TLR) intracellular signaling pathway, which is open to modification by interleukin-1 receptor-associated kinase (IRAK) inhibitors, monoclonal antibodies, oligonucleotides, lipid-A analogs and microRNA.
Ibrutinib and IRAK4 inhibitors demonstrated better antitumor activity against CLL cells in vitro when used in combination than when each agent was used alone [159,160]. The IRAK4 inhibitor CA-4948 is currently under investigation in R/R CLL and other indolent B-cell malignancies as a single agent or in combination with ibrutinib (NCT03328078). Other possible targets for CLL drugs have also been identified recently. For example, the ERK inhibitor ulixertinib decreased ERK phosphorylation in MAPK-mutated CLL cells. The MEK1/2 inhibitor binimetinib also inhibited CLL survival induced by stroma-conditioned media and phorbol myristylate (PMA), when used alone or in combination with venetoclax [161]. The combination of these agents with BTKi may be an effective treatment strategy for CLL. Finally, the SYK inhibitor entospletinib also appears to be a promising drug with clinical activity for patients previously treated with BCR-pathway inhibitors; treatment was found to yield OS of 32.7% and PFS of 5.6 months when used in monotherapy in patients with R/R CLL who had been previously treated with ibrutinib, spebrutinib, idelalisib or umbralisib [162].
Treatment options in patients resistant to novel agents like BTKi, PI3k inhibitors and venetoclax are limited [163]. Cellular treatment options like allogeneic hematopoietic stem cell transplantation (allo HCT), chimeric antigen receptor (CAR) T-cells and CAR NK-cells appear promising, especially in patients who failed BTKi combinations [164]. The development of more specific targeted therapies and novel combination treatments will help to design personalized effective treatments and strategies to improve the outcome of patients with CLL
Again in terms of efficacy of BTK inhibitors and patient’s monitoring, and having in mind the “future directions”, it would be nice to discuss the model of temporal dinamics upon Ibrutinib treatment in CLL made by single-cell immune profiling (Nat Commun. 11:577, 2020) as an innovative frontier for the management of these patients.
Response: This issue is added for Future directions chapter and discussed.
Specific comments
A figure depicting the kinome profile of the up-to-date approved BTK inhibitors would be helpful to the reader to immediately get the message of potency and selectivity of different compounds. As you know, figures provide added value to the piece, especially in review articles.
Response: The proper figure is added as Fig. 1.
It has been extensively demonstrated that in CLL the Src kinase Lyn plays the pivotal role in the pathogenesis of the diseases, mainly by its constitutive phosphorylation. Please correct/complete the sentence on page 3 lines 86-88.
Response: The sentence “It has been also demonstrated that in CLL the Src kinase Lyn plays the pivotal role in the pathogenesis of the diseases, mainly by its constitutive phosphorylation.” has been added
Tables 2 and 3 summarize the most important clinical trials. However, to better compare the studies, I suggest to add i) the median follow-up of each study (which is pretty different among studies), ii) Rab and OS as for example 3 years PFS, iii) discontinuation rates.
Response: The tables 2 and 3 has been modified according to the Reviewer suggestion. the median follow-up, discontinuation rates and p values for PFS and OS have been added.
Why did the authors decide to exclude reporting studies with ibrutinib + venetoclax? Authors should explain this choice, since this fixed duration combination will become relevant in next future. Studies with ibrutinib plus ublituximab are also missing.
Response: Studies with ibrutinib + venetoclax are included and studies and ibrutinib plus ublituximab are added.
In the GLOW open-label, randomized phase 3 study, 106 previously-untreated patients were randomized to receive ibrutinib plus venetoclax and 105 to receive chlorambucil plus obinutuzumab (Table 3) [48]. Patients with del(17p) or TP53 mutations were excluded. The participants received 420 mg/day ibrutinib for three months, followed by 12 cycles of ibrutinib plus venetoclax. In the control arm, the patients received six cycles of chlorambucil plus obinutuzumab at standard doses. The independent reviev commitee (IRC) found ORR values of 86.6% for the ibrutinib-venetoclax arm and 84.8% for the chlorambucil-obinutuzumab arm. However, the ibrutinib-venetoclax arm yielded a significantly higher CR rate (38.7%) than chlorambucil-obinutuzumab (11.4%) (p<0.0001). The undetectable MRD in BM scores at three months following end of treatment were 51.9% and 17.1%, respectively (p<0.0001). With a median follow-up of 27.7 (range, 1.7 to 33.8) months, median IRC-assessed PFS was not reached for the ibrutinib-venetoclax arm and 21.0 months for the chlorambucil-obinutuzumab arm; estimated 24-month PFS rates were 84.4% versus 44.1%, respectively. In addition, 90% of responders to ibrutinub-venetoclax demonstrated maintained response after two years compared with 41% of responders to chlorambucil-obinutuzumab.
Time-limited treatment with acalabrutinib, venetoclax, and obinutuzumab was also investigated in patients with previously-untreated CLL in a phase 2 study [57]. Therapy consisted of acalabrutinib alone for cycle 1, and acalabrutinib with obinutuzumab for six cycles. Venetoclax was used from the beginning of cycle 4 until day 1 of cycle 16 or day 1 of cycle 25. Patients with CR and undetectable MRD in the BM could discontinue therapy at the start of cycle 16 or at the start of cycle 25 if they were at least in PR. Undetectable MRD was defined as less than 1 CLL cell per 10 000 leucocytes, as measured by four-colour flow cytometry. At cycle 16 day 1, 14 (38%) of 37 patients obtained CR with undetectable MRD in the BM. An ongoing phase 3 study (NCT03836261) is currently comparing the effects of combined acalabrutinib and venetoclax, with and without obinutuzumab, with the investigator's choice of chemoimmunotherapy in previously untreated CLL.
Page 10, line 406. Data from the SEQUOIA trial arm D (Zanobrutinib plus Venetoclax) have recently been presented at the last ASH meeting.
Response: The SEQUOIA Arm D results are included
Early results for the SEQUOIA trial (NCT03336333), based on patients with treatment-naive del(17p) CLL/SLL receiving zanubrutinib + venetoclax in Arm D, has been recently presented at the American Society of Hematology meeting [68]. With a median follow-up of 9.7 months, 35 patients were included and 32 patients remained on treatment. Zanubrutinib combined with venetoclax was well tolerated and no new safety signals were identified. For the 31 patients who reached the initial efficacy assessment at three months after starting zanubrutinib, 30 (96.8%) responded to treatment and one patient had disease progression
Why did the Authors take Evobrutinib, Elsobrutinib, Tolebrutinib, Rilzabrutinib into account ? They are under investigation for immune-mediated diseases and data on CLL are not available. I suggest to remove them, thus getting room to face the problems reported above.
Response: Evobrutinib, Elsobrutinib, Tolebrutinib, Rilzabrutinib are removed. However, SHR1459 (TG-1701) and DTRMWXHS-12 ( DTRM-12) are add as they are investigated in CLL
3.4.4. SHR1459
SHR1459 (TG 1701, EBI-1459; Reistone Biopharma,. Jiangsu Hengrui Medicine Co.) is a second-generation, covalently-bound and irreversible second-generation BTKi currently under clinical development. This agent has been found to demonstrate superior selectivity to BTK compared to ibrutinib in in vitro kinase screening [80]. SHR1459 therapy, alone or in combination with ublituximab and umbralisib, is currently under clinical development in phase 1 trial in patients with R/R mature B cell neoplasms or CLL (NCT03671590; NCT04806035) [81].
3.4.5. DTRMWXHS-12
DTRMWXHS-12 (DTRM-12) (NCT02900716) is a pyrazolo-pyrimidine derivative irreversible BTK inhibitor currently under phase 1 and phase 2 clinical trials for CLL and NHL [82]. DTRMWXHS-12, used alone and in combination with everolimus and pomalidomide, has yielded encouraging findings in several high-risk, multirefractory CLL and NHL patients, including those previously treated with ibrutinib, in simultaneous phase I studies [83]. DTRM-12 monotherapy was well tolerated across B cell malignancies and CLL in both studies. No dose-limiting toxicity (DLT) was observed and MTD was not identified. PK studies demonstrate adequate target drug exposures at all dose levels. A phase II expansion cohort study of DTRMWXHS-12 in combination with everolimus and pomalidomide in patients with refractory or relapsed CLL and NHL is ongoing (NCT04305444).
In the paragraph dealing with cardiovascular events and infections, I suggest to summarize the most important ibrutinib-induced atrial fibrillation scores as well as the issue of ibrutinib-induced infections (see Cancers 13:3240,2021).
Response: Tis part of the paper has been reduced and reedited.
Minor Points
- Please replace IgVH with IGHV gene.
Response: Corrected accordingly.
- Page 7 line 236. I think that the reference Table is #1 rather than Table #2.
Response: Corrected accordingly.
Reviewer 2 Report
Tadeusz Robak et al. uncovered BTK in CLL.
Points to be considered:
1) The rationale of why the authors came up with this review.
2) What is the information that is not exactly available that motivated the authors to come up with this information. What are the current caveats and how do the authors highlight the current research in answering them? If not they need to address in future directions.
3) A synergistic effects of BTK and CD20- based therapy can be seen on inducing apoptosis of tumor cells (comment on that)
4)In the frame of point 3 thinking, (anti-CD20) emerged as the standard of care in the induction treatment of chronic lymphocytic leukemia (CLL). Modern therapy can still be inspired by the authors findings and by opportunities offered by bispecific antibodies and antibody-associated immune modulation are addressed (refer to PMID: 26818572)
5)this reviewer personally misses a general figure/graphical abstract referring to pathophysiological pathways and related theragnostic strategies
6) The underlying message here is that more precision and individualized approaches need to be tested in well designed clinical trials – a challenge, but I would be interested in their perspective of how this might be done.
7) The authors need to highlight what new information the review is providing to enhance the research in progress.
Author Response
Reviewer 2.
The rationale of why the authors came up with this review.
What is the information that is not exactly available that motivated the authors to come up with this information. What are the current caveats and how do the authors highlight the current research in answering them? If not they need to address in future directions.
Response: Response: This is an invited paper and the topic suggested by the editor. However the rationale is added in the end of introduction
Considerable efforts are underway to reduce the toxicity of CLL treatments and improve their efficacy, which have resulted in the design of a number of new agents. However, while these agents have shown encouraging results in early clinical results, no long-term data is yet available. This review therefore summarizes the pharmacology, clinical efficacy, safety, dosing and drug-drug interactions of BTKi in the treatment of CLL, and discusses its further implications. It includes the most recent results from the ongoing clinical trials and preclinical studies. It also presents novel concepts for the management of CLL, including the use of optimal drug combinations, novel irreversible and reversible BTKis and more precise and individualized approaches intended to enhance the progress and development of well-designed clinical trials.
A synergistic effects of BTK and CD20- based therapy can be seen on inducing apoptosis of tumor cells (comment on that)
Response: A synergistic effects of BTK and CD20 antibodies are commented.
BTK inhibitors can be combined with other targeted agents known to be active in CLL. They have demonstrated synergistic effects with anti-CD20 monoclonal antibodies in inducing apoptosis in tumor cells [26,27]. In addition, the combination of ibrutinib and a BCL-2 antagonist showed additive or more than additive cytotoxicity in vitro against CLL cells from patients treated with ibrutinib [28]. These findings suggest that combinations of BTK inhibitors, BCL-2 antagonists and/or anti-CD20 monoclonal antibodies should be tested clinically against CLL, to increase antileukemic efficacy and reduce the risk of acquired resistance. The mechanism of action of BTK inhibitors is summarized in Fig. 1.
In the frame of point 3 thinking, (anti-CD20) emerged as the standard of care in the induction treatment of chronic lymphocytic leukemia (CLL). Modern therapy can still be inspired by the authors findings and by opportunities offered by bispecific antibodies and antibody-associated immune modulation are addressed (refer to PMID: 26818572)
Response: Comments on bispecific antibodies and antibody-associated immune modulation is added in Future directions
- Future directions
The BTK inhibitors ibrutinib and acalabrutinib, and venetoclax, induce long-lasting remissions in most patients with CLL, with or without CD20 antibodies. However, their use may eventually be associated with the development of clinical resistance or unacceptable toxicity. When used as single drugs in CLL, BTK inhibitors are given continuously until disease progression or unacceptable toxicity. BTKi combinations is given for a limited time [154]. In addition, BTKi resistance can be overcome and toxicity can be reduced with the use of combination therapies. Several ongoing studies are therefore focusing on time-limited combination therapy strategies with incorporate venetoclax and CD20 monoclonal antibodies. Such combination therapy may also elicit a deeper response. However, the toxicity, financial costs and efficacy of the drug should be taken into account when deciding on the optimal therapeutic solution.
Hope in the treatment of CLL resistant for currently-available drugs is offered by combining BTKi with modern antibody-based therapies [155]. These include bispecific antibodies, antibody-drug conjugates, antibody-associated immune modulation and other agents. Bispecific antibodies have shown antileukemic activity against CLL cells in vitro and in a xenograft mouse model. The bispecific anti-CD3 and anti-CD20 antibody epcoritamab is currently under evaluatin in a phase 1 trial for R/R CLL [NCT04623541]. In response to clinical and pharmacological considerations, ibrutinib therapy has been tested in combination with the proteasome inhibitor carfilzomib and the HDAC inhibitor abexinostat in preclinical models, and initial clinical trials for such ibrutinib-based combination therapies are ongoing [156,157].
Another option is the introduction of novel, more specific and less toxic BTKi agents that can be more effective and safer than ibrutinib and second generation BTKis. The next generation of reversible BTKis bind non-covalently to BTK, but are still active in CLL cells harboring the most common resistance mutations (BTK C481S). Two third-generation reversible BTKis, pirtobrutinib and nemtabrutinib, yielded promising results in early clinical trials and are now under examination in phase 2 and phase 3 trials (NCT05023980, NCT04965493, NCT04728893, NCT03162536).
Recently, other genetic alterations have been identified in CLL that may play a role in the prognosis and evolution of CLL, including those associated with the TLR, MAPK and Notch signaling pathways [158]. However, no information is currently available regarding the efficacy of novel targeted drugs, including BTK inhibitors, in patients with recurrent mutations other than TP53 and NOTCH1. Alternatively, another target for potentially useful novel agents in CLL is the toll-like receptor (TLR) intracellular signaling pathway, which is open to modification by interleukin-1 receptor-associated kinase (IRAK) inhibitors, monoclonal antibodies, oligonucleotides, lipid-A analogs and microRNA.
Ibrutinib and IRAK4 inhibitors demonstrated better antitumor activity against CLL cells in vitro when used in combination than when each agent was used alone [159,160]. The IRAK4 inhibitor CA-4948 is currently under investigation in R/R CLL and other indolent B-cell malignancies as a single agent or in combination with ibrutinib (NCT03328078). Other possible targets for CLL drugs have also been identified recently. For example, the ERK inhibitor ulixertinib decreased ERK phosphorylation in MAPK-mutated CLL cells. The MEK1/2 inhibitor binimetinib also inhibited CLL survival induced by stroma-conditioned media and phorbol myristylate (PMA), when used alone or in combination with venetoclax [161]. The combination of these agents with BTKi may be an effective treatment strategy for CLL. Finally, the SYK inhibitor entospletinib also appears to be a promising drug with clinical activity for patients previously treated with BCR-pathway inhibitors; treatment was found to yield OS of 32.7% and PFS of 5.6 months when used in monotherapy in patients with R/R CLL who had been previously treated with ibrutinib, spebrutinib, idelalisib or umbralisib [162].
Treatment options in patients resistant to novel agents like BTKi, PI3k inhibitors and venetoclax are limited [163]. Cellular treatment options like allogeneic hematopoietic stem cell transplantation (allo HCT), chimeric antigen receptor (CAR) T-cells and CAR NK-cells appear promising, especially in patients who failed BTKi combinations [164]. The development of more specific targeted therapies and novel combination treatments will help to design personalized effective treatments and strategies to improve the outcome of patients with CLL
this reviewer personally misses a general figure/graphical abstract referring to pathophysiological pathways and related theragnostic strategies
Response: The figure is added as requested ( Figure 1)
The underlying message here is that more precision and individualized approaches need to be tested in well designed clinical trials – a challenge, but I would be interested in their perspective of how this might be done.
Response: Comment is added in the future directions
The authors need to highlight what new information the review is providing to enhance the research in progress.
Response: The information provided by the review are added in the end of introduction.
Considerable efforts are underway to reduce the toxicity of CLL treatments and improve their efficacy, which have resulted in the design of a number of new agents. However, while these agents have shown encouraging results in early clinical results, no long-term data is yet available. This review therefore summarizes the pharmacology, clinical efficacy, safety, dosing and drug-drug interactions of BTKi in the treatment of CLL, and discusses its further implications. It includes the most recent results from the ongoing clinical trials and preclinical studies. It also presents novel concepts for the management of CLL, including the use of optimal drug combinations, novel irreversible and reversible BTKis and more precise and individualized approaches intended to enhance the progress and development of well-designed clinical trials

Round 2
Reviewer 1 Report
My comments have been adequately addressed.
Reviewer 2 Report
The authors have clarified several of the questions I raised in my previous review. Most of the major problems have been addressed by this revision.